# In-cell structure and variability of pyrenoid Rubisco

Nadav Elad [1,2,7], Zhen Hou [3,7], Maud Dumoux [4], Alireza Ramezani [5], Juan R. Perilla [5] & Peijun Zhang [2,3,6] ✉

Ribulose-1,5-bisphosphate carboxylase/oxygenase (Rubisco) is central to global $CO_2$ fixation. In eukaryotic algae, its catalytic efficiency is enhanced through the pyrenoid - a protein-dense organelle within the chloroplast that concentrates $CO_2$. Although Rubisco structure has been extensively studied in vitro, its native structure, dynamics and interactions within the pyrenoid remain elusive. Here, we present the native Rubisco structure inside the green alga *Chlamydomonas reinhardtii* determined by cryo-electron tomography and subtomogram averaging of cryo-focused ion beam milled cells. Multiple structural subsets of Rubisco are identified, stochastically distributed throughout the pyrenoid. While Rubisco adopts an active conformation in the best-resolved map, comparison among the subsets reveals significant local variations at the active site, at the large subunit dimer interfaces, and at binding protein contact regions. These findings offer a comprehensive understanding of the structure, dynamics, and functional organization of native Rubisco within the pyrenoid, providing valuable insights into its critical role in $CO_2$ fixation.

In the effort to mitigate global warming, carbon fixation, as a part of photosynthesis, has come into the spotlight[1]. Additionally, the ever-growing global population places a formidable burden on the sustainable food supply, which is closely linked to the efficiency of carbon fixation[1–3]. As the most abundant protein, ribulose-1,5-bisphosphate carboxylase-oxygenase, commonly known as Rubisco, is indispensable for life on Earth as it catalyzes the reaction between $CO_2$ and ribulose-1,5-bisphosphate (RuBP), resulting in the production of two molecules of 3-phosphoglycerate (3-PGA) in the Calvin-Benson cycle[4,5]. Despite its crucial role, Rubisco is known for its relatively slow catalytic rate and its dual activity, meaning it can also react with $O_2$ instead of $CO_2$, which in turn reduces the efficiency of carbon fixation[5–8].

The structure of Rubisco is highly conserved across different species, reflecting its essential role in carbon fixation[9]. Rubisco is a holoenzyme, typically composed of two types of subunits. In higher plants, algae, and cyanobacteria, the enzyme is a hexadecamer, consisting of eight large subunits and eight small subunits[5,10–15]. The large subunits form the core of the enzyme where the catalytic sites are located, while the small subunits play a regulatory role, influencing the enzyme's activity and stability[10,16]. Numerous structures of purified Rubisco have revealed the catalytic cycle in detail[10,17]. The active site is located at the interface between the C-terminal of one large subunit and the N-terminal of an adjacent one, such that two large subunits form a functional dimer that harbors two active sites (see Fig. 2b below). A rigid framework is formed around the active site, while several mobile loops, including the N and C termini and the important loop 6, mediate the opening and closing of the substrates binding pocket[10,18–20]. Yet, it remains to be seen whether the structural patterns and dynamics that have been revealed in vitro, are maintained in the crowded environment of the cell, and

[1]Department of Chemical Research Support, Weizmann Institute of Science, Rehovot, Israel. [2]Diamond Light Source, Harwell Science and Innovation Campus, Didcot, UK. [3]Division of Structural Biology, Nuffield Department of Medicine, University of Oxford, Oxford, UK. [4]The Rosalind Franklin Institute, Harwell Science & Innovation Campus, Didcot, UK. [5]Department of Chemistry and Biochemistry, University of Delaware, Newark, DE, USA. [6]Chinese Academy of Medical Sciences Oxford Institute, University of Oxford, Oxford, UK. [7]These authors contributed equally: Nadav Elad, Zhen Hou. ✉e-mail: peijun.zhang@strubi.ox.ac.uk

how the cellular Rubisco population varies with respect to catalytic activity.

In the unicellular green alga *Chlamydomonas reinhardtii*, Rubisco is sequestered within a specialized non-membrane-bound organelle called the pyrenoid, located in the chloroplast[21,22]. One-third of the global $CO_2$ is arguably fixed by pyrenoid Rubisco[23]. The pyrenoid is primarily found in many eukaryotic algae[21,22,24,25] and plays a crucial role in enhancing the efficiency of carbon fixation, a process believed to have been driven by the gradual decrease of atmospheric $CO_2$ over billions of years until the recent industrial revolution by humans[26]. This compartmentalization is particularly advantageous under low $CO_2$ conditions, as it helps to concentrate $CO_2$ around Rubisco through $CO_2$-concentrating mechanisms (CCM), minimizing the enzyme's oxygenase activity and reducing photorespiration[27–29].

The concentration of Rubisco in the pyrenoid of *C. reinhardtii* is mediated by the protein Essential Pyrenoid Component 1 (EPYC1), forming liquid-liquid phase separation (LLPS)[12,23,30]. Unlike its analogues in cyanobacteria: the linker protein CsoS2 in α-carboxysomes and CcmM in β-carboxysomes, which bind to the interface of two large subunits of Rubisco[13,14,31], EPYC1 binds to the small subunit directly[12]. Recent studies have demonstrated intriguing packaging patterns for Rubisco in α- and β-carboxysomes, which are confined by shells[14,32–35]. Whether similar or distinguished packaging patterns exist for Rubisco in the pyrenoid remains to be explored. Pioneering in-cell cryo-electron tomography (cryo-ET) works on the *C. reinhardtii* pyrenoid have suggested potential packing patterns for Rubisco, revealed its tight connection with thylakoids and presented a single in-cell structure of Rubisco at 16.5 Å[12,21,36]. However, higher-resolution in-cell structures of Rubisco and more detailed spatial analyses of Rubisco in the pyrenoid are essential to build a robust and comprehensive model that fully captures the complexity and dynamic nature of this organelle and its critical role in life-sustaining processes.

In this work, we employ cryo-focused ion beam (cryo-FIB) milling and cryo-ET, couple with subtomogram averaging (STA) to investigate the pyrenoid and Rubisco in *C. reinhardtii*. We obtain an in-cell structure of Rubisco at sub-nanometer resolution, enabling us to perform molecular dynamic flexible fitting (MDFF), thereby revealing a closed conformation of Rubisco in the cell. Successive analyses show localized heterogeneity in Rubisco conformations and pyrenoid binding partners, reflecting its dynamic nature and enzymatic activity in carbon fixation. Moreover, the distribution of Rubisco in the pyrenoid is revealed to be overall stochastic with local clusters showing specific arrangements, different from the rigid packaging in the α- and β-carboxysomes, and some variations in the distribution of Rubisco classes relative to their locations in the pyrenoid. Taken together, our work enhances the understanding of the pyrenoid and Rubisco in intact cells, providing valuable insights for future studies on fundamental mechanisms and the bioengineering of pyrenoid and Rubisco in other organisms.

## Results

### Structural heterogeneity of pyrenoid Rubisco is revealed by the classification of sub-volumes

To gain insight into the structure of pyrenoid Rubisco, we vitrified *C. reinhardtii* cells on TEM grids and used cryo-FIB to cut thin lamellae through the cells. We then applied cryo-ET to collect tilt series from lamellae at pyrenoid areas (Supplementary Tables 1, 2). Rubisco complexes are highly abundant within the pyrenoid and appear in their expected spherical morphology (Fig. 1a). Additionally, thylakoid, pyrenoid, and mini-tubules can be clearly seen (Fig. 1a–c, Supplementary Movie 1). To resolve the structure of Rubisco, deep learning-based particle picking was performed, and most discernable Rubisco particles were correctly picked (Fig. 1c, Supplementary Fig. 1). In the following STA analysis, iterative 3D classification and refinements were applied. Interestingly, even though Rubisco particles were

ubiquitously present in tomograms and seemed to have similar overall morphology, iterative 3D classifications revealed vast heterogeneity at medium-low resolutions (Supplementary Fig. 2, Supplementary Table 3, and see below). Experimenting with different 3D classification strategies, we found that classification and cleaning the data from "junk" particles without applying symmetry gave more reliable results compared to applying D4 symmetry from the start, and classifying into 20 classes yielded the best results. This likely represents a balance between the vast heterogeneity in the dataset and the number of images available per class. Attempts to classify into different numbers of classes, join, or sub-classify the resulting 20 classes, did not improve resolutions of the reconstructions. "Good" 3D classes were selected based on the overall appearance of Rubisco features, the map's resolution, and accuracy in the angular assignment. Following classification with no applied symmetry, classes were refined separately with D4 symmetry applied and finer bin sizes to resolutions of 8 to 15 Å. We start by describing the best-resolved Rubisco class and the associated secondary structure model.

### The best-resolved Rubisco complex is in a closed conformation state

The best resolved 3D class (number 12) was refined to 8.1 Å resolution, enabling the identification of secondary structure elements (Fig. 1d, Supplementary Fig. 3, Supplementary Table 4). The map contains 12% of the good particles (i.e. 148,639 particles contained in the C1-refined classes, Supplementary Fig. 2f, Supplementary Table 3). To better understand the reconstructed conformational state, we used molecular dynamics flexible fitting (MDFF, detailed in the methods section) to refine the published X-ray coordinates from purified *C. reinhardtii* Rubisco complexes within our map (Fig. 2a). Even though side chains cannot be resolved at this resolution, secondary structure elements from the x-ray structures can be modeled reliably, in conjunction with known protein structural constraints. We then compared the refined coordinates to published structures of known functional states. Overlaying the class 12 MDFF coordinates with those of PDB 1GK8[15] resulted in a root-mean-square distance (RMSD) of 2.75 Å. An excellent match is seen at the center of the complex near the 2-fold axis and at the interior-facing parts of the large subunit (Fig. 2b). The MDFF structure is more extended along the 4-fold axis compared to the 1GK8 structure, producing mismatches of up to about 4 Å between corresponding helices at the small subunits and nearby large subunit segments.

However, rather than major whole-domain movements, the transition between open (inactivated) and closed (activated) states has been shown to involve significant local conformational changes of loops around the active site. In particular, the large subunit loop 6 (aa 331–338) and C-terminal (aa 461–475) cover the active site in the closed conformation, and both fold outwards and become disordered in the open conformation. Loop 64-68 and the large subunit N-terminal (aa 7–21) cover the other side of the active site in the closed conformation and are disordered in the open conformation. We compared these elements between class 12 MDFF model and either 1GK8[15] or 1EJ7[20] coordinates, representing a fully closed or fully open conformation, respectively (Fig. 2c). The conformations of these four elements in our model clearly match the closed-state conformation as they overlay well with the 1GK8 structure. Most strikingly, the positions of mobile loops 6 and 46–68 clearly cover the location of the active site, and map density is observed at this position up to high contour values (Supplementary Fig. 4a–c). The map features density which correlates well with the expected closed conformations of the C and N termini, although these are fully present at lower sigma values. This may indicate a higher level of heterogeneity or can be due to their peripheral locations. As seen in Supplementary Fig. 4d–f, the N terminus is well accounted for in the map, while the C-terminus density is interrupted in the middle,

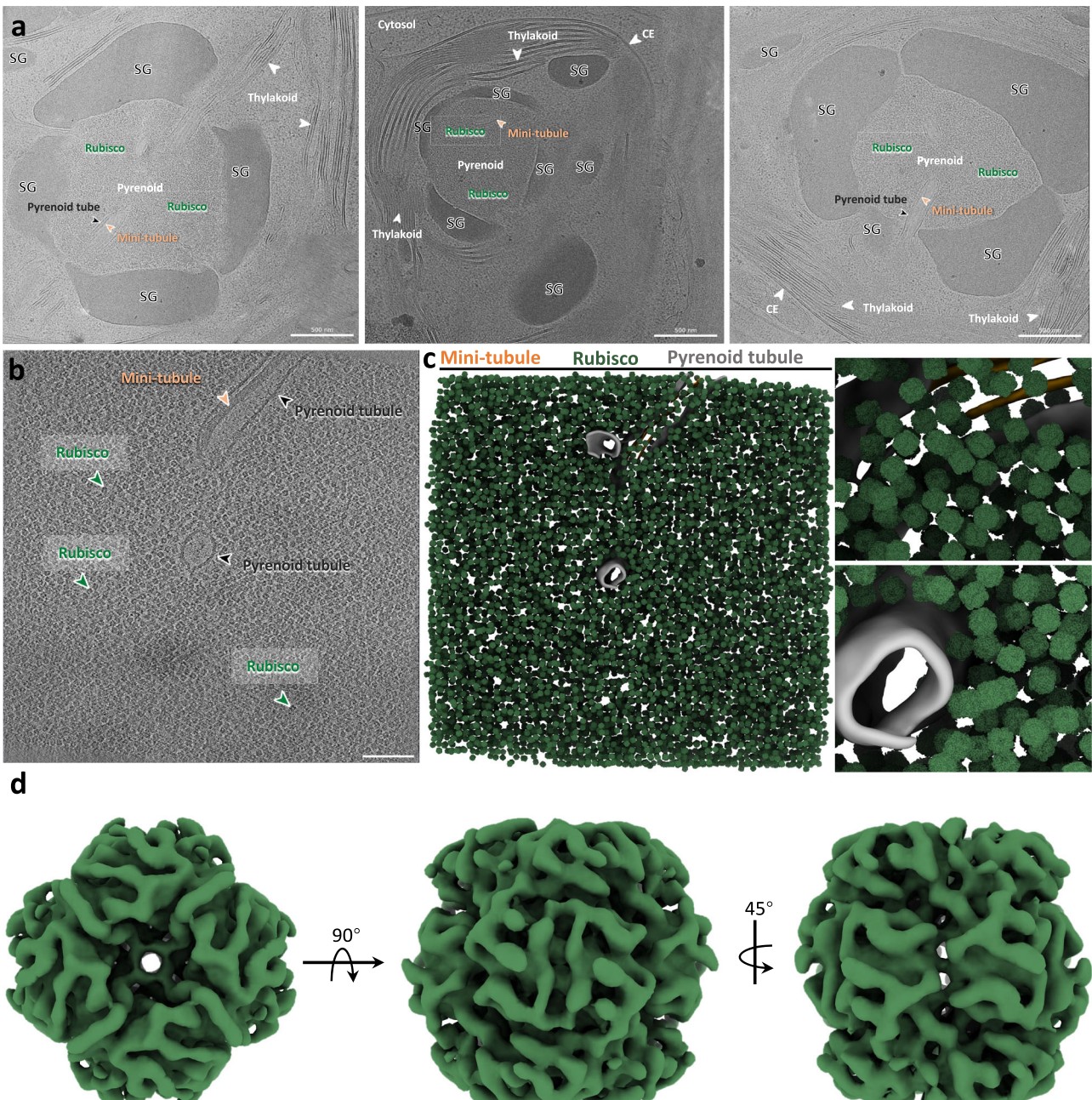

**Fig. 1 | Cryo-ET of pyrenoid and in-cell structure of Rubisco. a** Overviews of phase-separated Rubisco in intact pyrenoids of various sizes. Cellular components are labelled and annotated accordingly, SG starch granule, CE chloroplast envelope. **b** A representative tomographic slice of the pyrenoid in *C. reinhardtii*. Thylakoid, pyrenoid tubule, and Rubisco particles are labeled and indicated accordingly. Scale bar = 100 nm. **c** The segmented volume of the tomogram in (**b**). The left panel shows an overview of the segmented pyrenoid with mapped-back Rubisco particles, the right panel shows two close-up views of Rubisco particles with the thylakoid. Pyrenoid tubule, mini-tubule, and Rubisco are colored accordingly. The particles displayed are after "junk" removal by classification, as described in the methods section. Rubisco map used for display is of the D4 symmetric class 12 described below. **d** The in-cell structure of Rubisco (D4 symmetric class 12) in *C. reinhardtii* shown in three orthogonal views (*n* = 17,713 particles, *n* = 26 tomograms).

around Glu468. Indeed, the C-terminus is known to be poorly anchored to the body of the large subunit even in the closed conformation, except for a network of interactions formed at the "latch site" by the conserved Asp473. This interaction stabilizes the closed active site and is important for catalytic efficiency and specificity[20,37]. The MDFF model positioned the end of the C-terminus in contact with the body of the complex within a strong peripheral density, correlating well with the location of the latch site (Supplementary Fig. 4e, f). This density is therefore more likely to be part of the C-terminus than a binding protein.

## Local conformational changes and binding proteins are the main sources of heterogeneity

Processing of a large dataset of pyrenoid Rubisco subtomograms has revealed the presence of multiple distinct classes of complexes (Supplementary Fig. 2). Exploring the origins of this heterogeneity is particularly intriguing, as the dataset captures authentic frozen snapshots of functional Rubisco, representing its dynamics and interactions with binding partners in the cell. Varying sources of heterogeneity can exist in such a cellular environment, including concerted whole-domain movements, local conformational variations, disruption of symmetry,

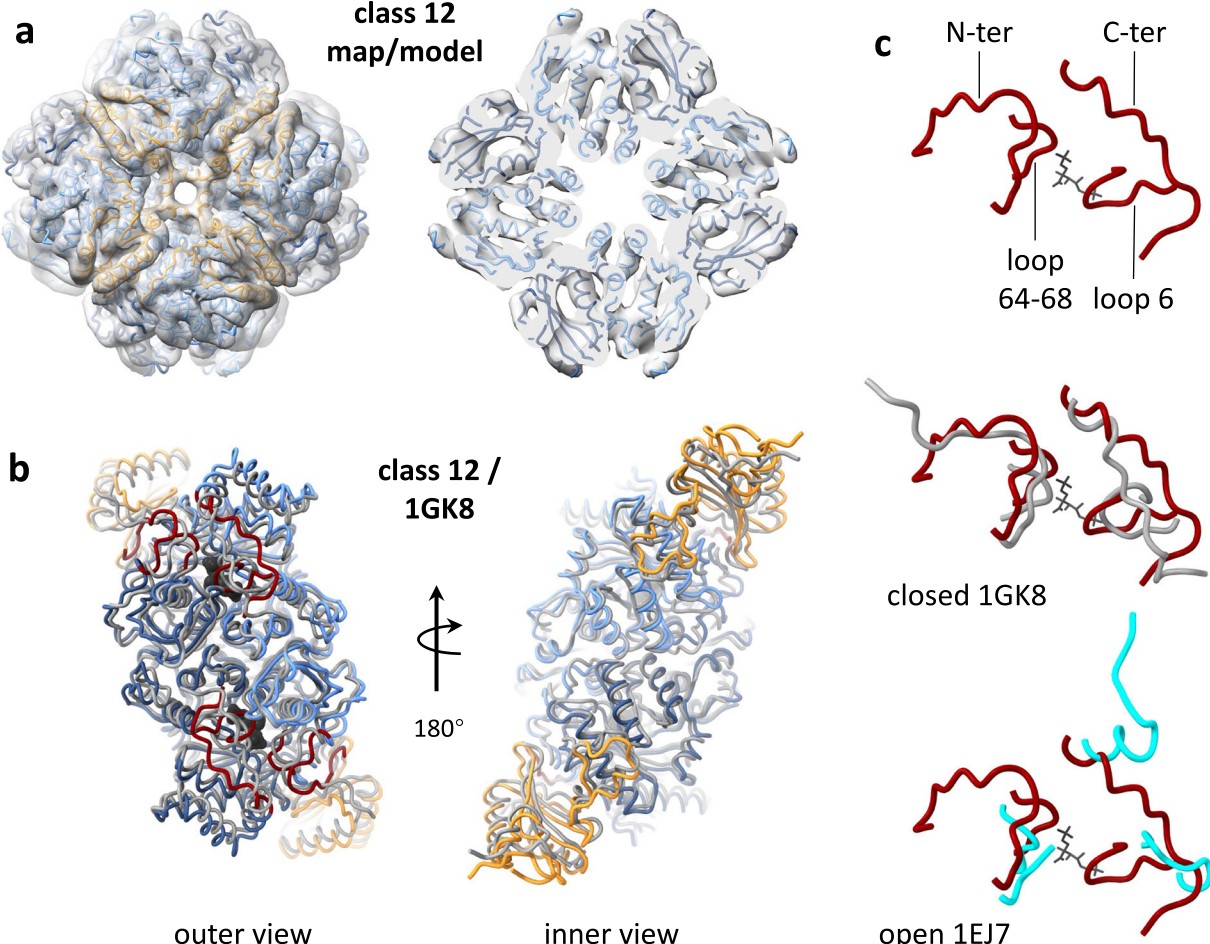

**Fig. 2 | Structure of best-resolved STA class. a** Map and fitted MDFF model of class 12, which refined to the highest resolution (8.1 Å). Shown are top and cut-away views. Large and small subunits are colored blue and orange, respectively. **b** Alignment between two large subunits and two small subunits of class 12 MDFF model and PDB 1GK8. The MDFF model is colored as follows: the top and bottom large subunits are colored light and dark blue, respectively; the small subunits are colored orange, and key active site segments are colored dark red. Key segments include the large subunit loop 6 (aa 331-338), C-terminus (aa 461-475), loop 64-68, and the N-terminus (aa 7–21). Corresponding chains from PDB 1GK8 are colored

grey. The 2-carboxyarabinitol-1,5-bisphosphate (2CABP) inhibitor from 1GK8, which marks the position of the active site, is shown in black atomic view. **c** Comparison of active site conformations. The conformational changes in these residues correlate with the Rubisco activity state. The MDFF model (dark red) is compared to PDB 1GK8 (grey, middle panel) and PDB 1EJ7 (cyan, bottom panel), which represent closed and open conformations, respectively. A better match is found for the closed state. The 2CABP inhibitor from 1GK8 is shown in black atomic view in all panels for reference.

and the presence of binding proteins. Yet, significant heterogeneity essentially limits the attainable resolution because it compromises the number of particles per class and, consequently, reduces the signal-to-noise ratio (SNR) in the average map.

We first compared the extent of whole domain movements between classes. To this end, we fitted the Rubisco coordinates into the three D4-symmetric classes that were refined to the highest resolution (classes 7, 10 and 13, comprising 6%, 8% and 5% of the good particles, respectively. Supplementary Table 3, Supplementary Fig. 3). The Class 12 MDFF model (described above, Fig. 2a) was used as a starting model for rigid body fitting of whole domains, and as a reference for comparison. As seen in Supplementary Fig. 5, only subtle whole-domain shifts of up to 2.5 Å occur between the class 12 model and the models fitted into all three classes. Although the extent of the observed movements is much smaller than the resolution of the maps, this result is in accordance with x-ray structures showing that Rubisco activity cycle and varying conditions induce only minor, angstrom-scale, whole-domain movement[10,19,20].

Next, we explored local variations in the maps as a source of heterogeneity between classes. Some local variations can be readily observed when comparing maps by visual inspection (Fig. 3a).

However, unlike class 12, the lower resolution obtained for the other classes restricted precise interpretation of their conformational state by flexible fitting. Nevertheless, local variations were tracked and highlighted by subtracting maps from one another and looking at the resulting difference maps (Fig. 3b–g). To this end, we calculated difference maps from confidence maps binarized at 1% false discovery rate threshold[38]. Interestingly, significant density gain is observed at the active when subtracting class 12 from each of the other three classes (7, 10 and 13), indicating that class 12 lacks density at this position relative to the three classes (Fig. 3b–d blue). As mentioned above (Fig. 2c) and shown before[18,39], the four large subunit segments - loop 6, the N- and C-termini, and loop 64–68, are particularly mobile during the catalytic cycle, shifting between a closed/active state and an open/inactive state, as well as populating intermediate states. The observation of significant density variation at this location suggests that the pyrenoid Rubisco population exists in multiple activity states. The extra density can also be attributed to active site binding proteins such as Rubisco activase[40] or neighboring Rubisco complexes, which bind in close proximity to the active site. Class 12, which is in the closed state, is particularly different from the rest of the classes at this position and perhaps constitutes a more stable conformation (Fig. 3b–d).

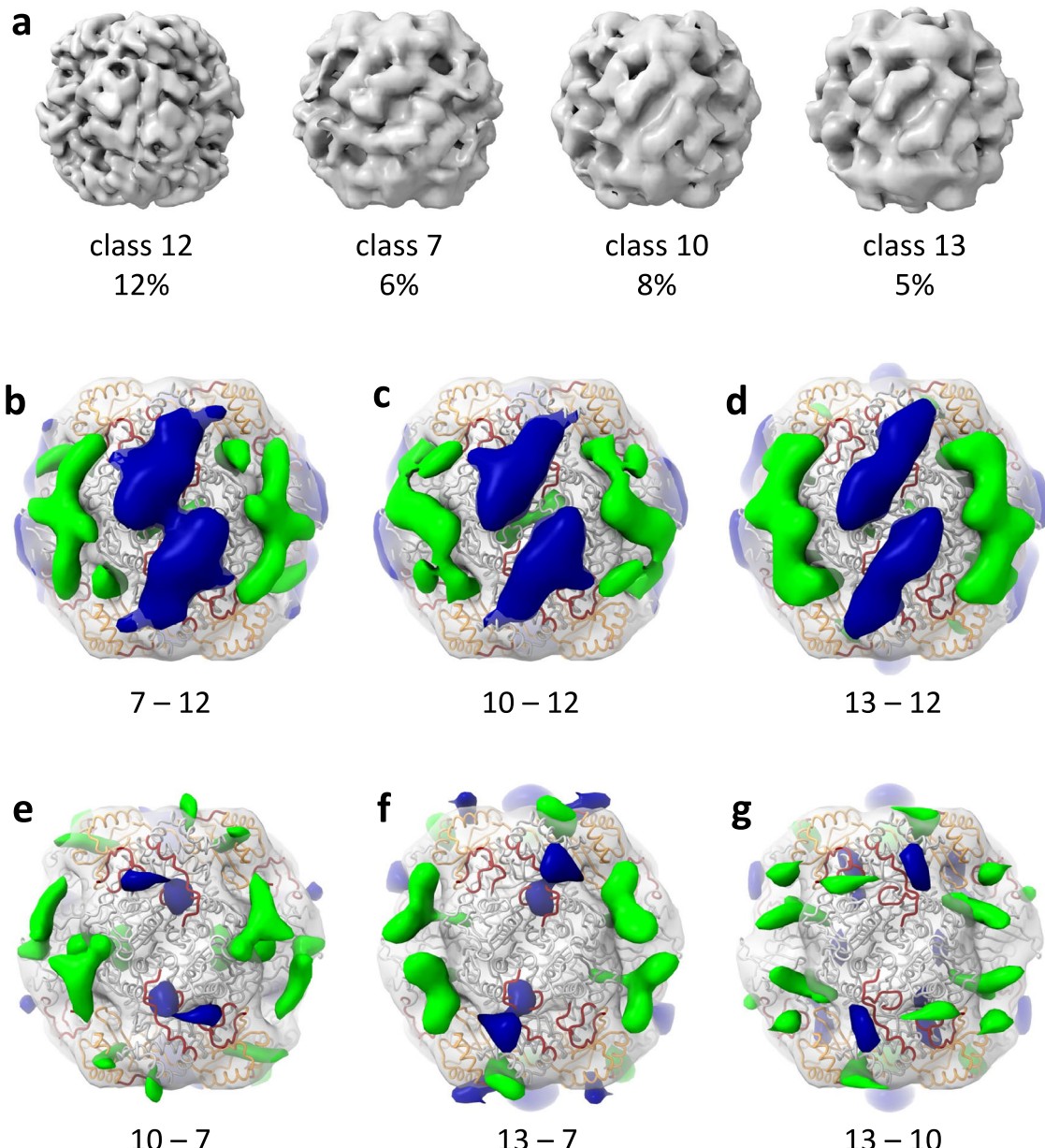

**Fig. 3 | Structural heterogeneity of pyrenoid Rubisco. a–c** D4-symmetric maps of four classes out of ten (Supplementary Fig. 2), which were refined to the highest resolution, along with their proportion in the "good" subtomogram dataset (Supplementary Table 3). **b–g** Difference maps showing major variations in density between classes. Positive densities resulting from the subtraction of one map from another as indicated in the panels are colored dark blue and negative densities are colored green. Difference maps between classes were calculated from confidence maps binarized at 1% false discovery rate threshold, Gaussian filtered, and presented at 10 standard deviations (SDs) contour level. For orientation, the difference maps are overlaid on the class 12 low-passed filtered map and fitted MDFF model (Fig. 2a). Large subunits, small subunits and active site residues are colored grey, orange, and dark red, respectively.

Smaller-scale differences at the active site can be seen between the other three classes (Fig. 3e–g).

Significant conformational variability is seen also between the large subunit dimers near the 2-fold axis. Particularly, class 12 shows extra density at this interface compared to all three classes (Fig. 3b–d, green). This difference density can be interpreted as correlated movement of subunits at the dimer interface. Additionally, significant difference densities protruding outside the Rubisco model appear next to the large subunit dimer interface when comparing classes 7, 10 and 13 (Fig. 3e–g, green). This position correlates well with the binding interface for carboxysome matrix proteins[13,14,31,33,35,41], and may indicate the presence of yet-unidentified proteins possessing novel modes of Rubisco binding within the pyrenoid, similar to those identified in

carboxysomes. Finally, difference densities are observed at several locations adjacent to the small subunits (Fig. 3e–g), including at the opening of the cavity along the 4-fold axis (Fig. 3d, f, g blue).

## Asymmetric binding to Rubisco

Functional pyrenoid Rubisco comprises 8 identical copies of the large and small subunits arranged in D4 symmetry. However, the Rubisco binding proteins may not bind to all subunits simultaneously[23,36], and conformational symmetry is not essentially maintained in the cellular context[42]. To explore the notion of asymmetricity in Pyrenoid Rubisco, we refined the sub-volume classes without applying D4 symmetry (Supplementary Fig. 2d, Supplementary Table 3). These C1 refinements compromised resolution compared to the corresponding

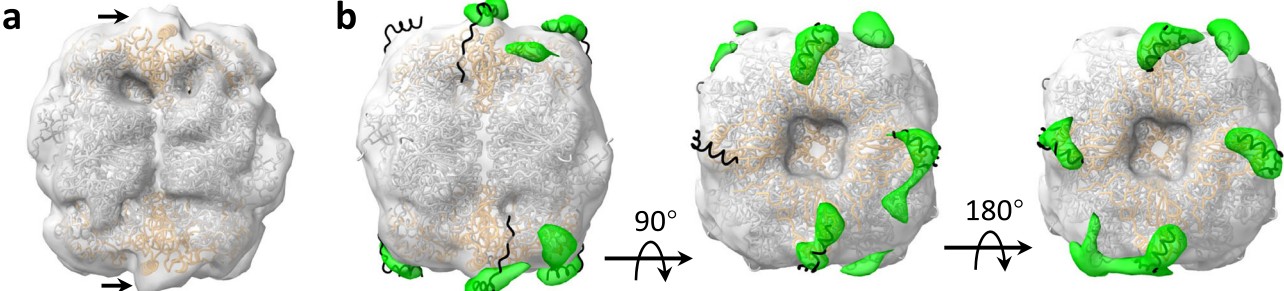

**Fig. 4 | Asymmetric binding of EPYC1 to Rubisco. a** Asymmetric map of class 12 with. The MDFF model from the D4 symmetric class 12 map was fitted as a rigid body (large and small subunits are colored grey and orange, respectively). Arrows indicate extra density next to the small subunits. **b** Difference density between the C1 and D4 symmetric class 12 maps (green), overlayed on the D4 symmetric class 12 map (low-passed filtered, transparent white). The difference map was calculated from confidence map binarized at 1% false discovery rate threshold, Gaussian filtered, and presented at 13 SDs contour level. The SPA structure of Rubisco-EPYC1 (PDB 7JFO) was fitted as a rigid body into the map, and the EPYC1 coordinates are colored black. The difference densities vary in shape and match the location of most EPYC1 helices.

D4 symmetric reconstructions. In particular, the best-resolved class 12 map was now refined to 13.1 Å, making the analysis of inter-subunit conformational heterogeneity impractical. Interestingly however, we observed that the C1 class 12 map presents a number of extra densities at the expected location of the EPYC1 binding site, next to the small subunits (Fig. 4a). EPYC1 was previously shown to have multiple alpha-helical repeats, which bind to the Rubisco small subunit, flanked by unstructured spacer regions[12,23]. Simultaneous binding of multiple Rubisco complexes supports its clustering in the pyrenoid.

The potential EPYC1-assiciated extra densities were not observed in the D4 symmetric class 12 map, likely because they were averaged out when applying symmetry. Thus, to validate the extra densities seen in the class 12 C1 reconstruction, we calculated a difference map between C1 and D4 maps, binarized at 1% false discovery rate threshold. As can be seen in Fig. 4b, the most significant difference in densities can be associated with the EPYC1 helix, since they are located next to the small subunits at the predicted binding site. These densities vary in shape and are absent next to one of the small subunits, indicating partial occupancy.

### Distribution of Rubisco in the pyrenoid

To investigate whether Rubisco particles are organized in a specific pattern within the pyrenoid as seen for Rubisco in α- and β-carboxysomes, we first analysed the distances and angles between the nearest neighbouring particles. The results revealed a largely random distribution, with a mean paired distance of ~13 nm (Fig. 5a) as previously observed[36]. This is consistent with the measured paired distances of 12.8 nm and 12.9 nm in two α-carboxysomes[14], but slightly larger than the distance in β-carboxysomes (12.2 nm)[33]. The pairwise average angle was measured ~90° (Fig. 5b). To explore potential patterns among Rubisco particles with restricted paired distances and angles, we categorized particles based on these parameters and mapped each category back to the tomograms. While no distinct global patterns emerged (Fig. 5c, d, Supplementary Fig. 6–8), local clusters were identifiable under stringent distance and angle constraints (Fig. 5e). Some fiber-like and spiral arrangements of the Rubisco were observed in these clusters (Fig. 5e, Supplementary Fig. 8a).

Given that STA analysis identified multiple classes of Rubisco in the pyrenoid, reflecting different conformational and binding states, we examined whether specific classes displayed preferential localization. We compared the distribution of the STA classes between $CO_2$-rich regions (defined as 200 nm from the surface of pyrenoid tubules[43]) and the rest of the pyrenoid. Intriguingly, the distribution of classes displayed differences compared with their overall distribution in the pyrenoid (Fig. 6a). Specifically, the best-resolved four classes showed lower abundance in the $CO_2$-rich regions. Following that, we

examined the pair-wise distances and angles between the nearest neighbours within and across the four best-resolved classes. The analysis indicated differences in the distance while the angular distribution was stochastic (Fig. 6b, Supplementary Fig. 8b, c). Class 10 showed the shortest distance to class 12 while class 13 deviated from the other three classes (Fig. 6b lower). The mapping of four best-resolved classes back to the tomograms suggested no obvious class-specific clustering (Fig. 6c). In summary, Rubisco particles in the pyrenoid exhibit a predominantly stochastic distribution, with short-range local clustering of some particles in the pyrenoid.

## Discussion

In the present study, we provide a detailed structural analysis of the Rubisco complex as it exists within the native pyrenoid. While the structure of Rubisco and its catalytic cycle have been thoroughly investigated in vitro using x-ray crystallography and single particle cryo-EM, these studies primarily relied on purified samples and focused on a single stable conformational state. Here, we employed cryo-FIB milling of *C. reinhardtii* pyrenoid sites in combination with cryo-ET, subtomogram averaging, and detailed 3D classification, allowing us to capture snapshots of functional Rubisco complexes in their native cellular environment, thereby avoiding the artifacts that can be involved in purification.

The vast heterogeneity present in the sample limits the achievable resolution, as it imposes classification into small subsets. Nevertheless, one of the classes was resolved at a resolution of 8.1 Å, allowing for accurate modeling of secondary structures. MDFF modeling indicated that this class features a Rubisco population in a closed conformation state, while comparison to other, lower resolution classes exposed the undelaying structural heterogeneity. Our study confirms that Rubisco does not exhibit significant whole-domain movements within the pyrenoid, consistent with findings from purified complexes[10]. The main variations between classes are characterized by local conformational changes, primarily around the active sites and the interface between large subunit dimers, while other significant variations are associated with locations and morphologies of binding proteins. The cores of the large and small subunits remain consistent, devoid of significant local conformational changes. Given this stability of the core, it is interesting to consider the factors that limit resolution. We suggest that several factors may contribute, including the high density of complexes within the pyrenoid, the gel-like matrix of unstructured proteins surrounding the Rubisco complexes, and sample thickness, all of which reduce SNR in the tomograms. Other factors that limit resolution may include asymmetries in catalysis between subunits[42] and variations in binding protein interactions, which were partially elucidated here.

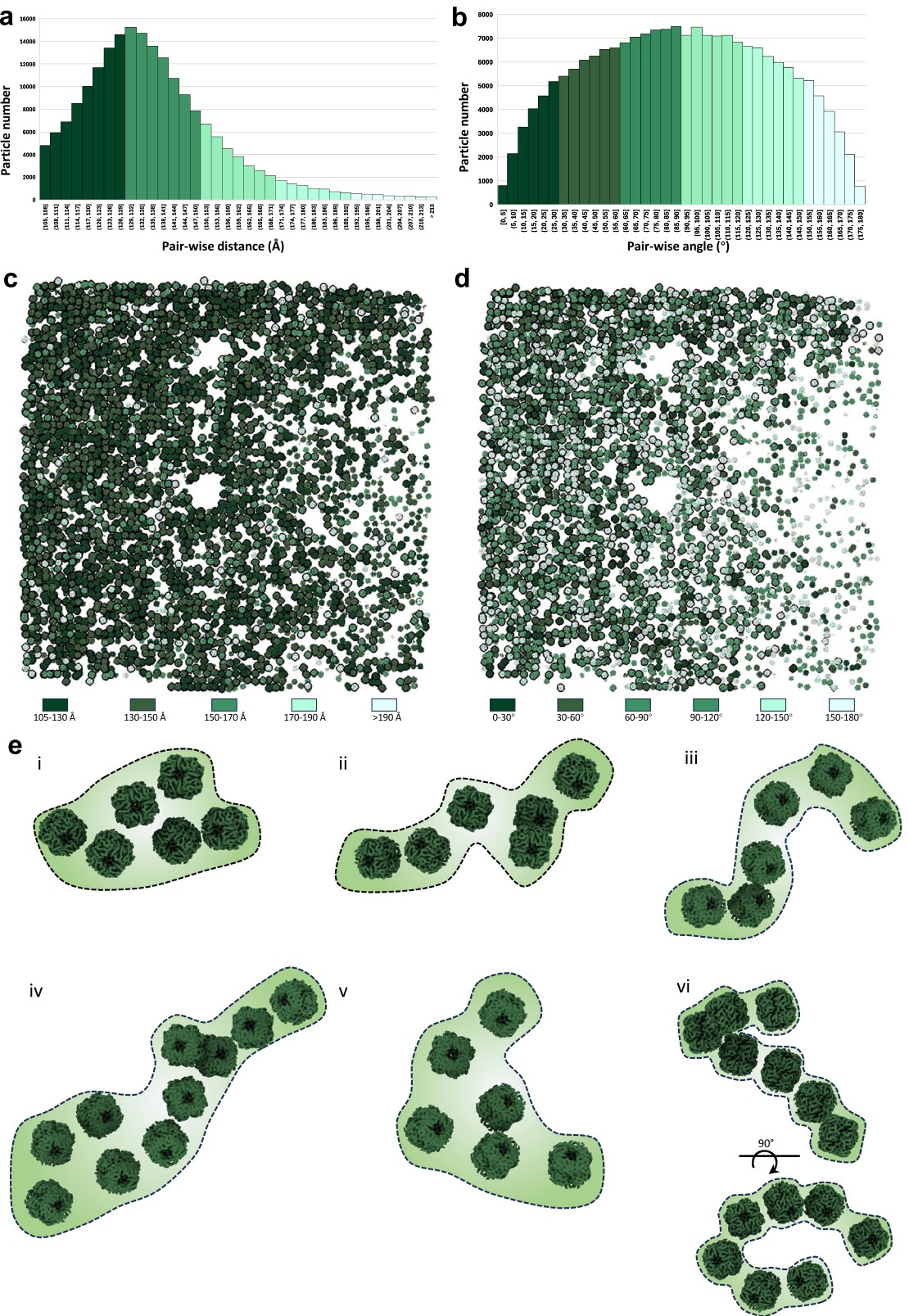

The identification of diverse structural patterns indicates that Rubisco complexes exhibit variations in their activity states and binding partners within the pyrenoid matrix. The work presents a snapshot of an active pyrenoid Rubisco population at variable stages of the catalytic turnover. Since *C. reinhardtii* Rubisco co-evolved with the pyrenoid, its kinetic parameters were adapted to the relatively high $CO_2$ concentration associated with the CCM. This mechanism promotes carboxylation and competitively inhibits oxygenation. Within the framework of kinetic trade-offs observed among Rubisco types, an association with the pyrenoid CCM generally promotes a higher carboxylation rate and weaker affinity for $CO_2$[44,45]. The reaction catalyzed by Rubisco consists of distinct steps and intermediates of variable stability. Substantial conformational shifts and flexibility at the active site facilitate the catalytic cycle, including substrate binding,

**Fig. 5 | Mapping-back of Rubisco in the pyrenoid. a** Distributions of pair-wise distances between the nearest Rubisco neighbours (from $n$ = 198,692, $n$ of tomograms = 26, particles with distance smaller than 10.5 nm were excluded). **b**, Distributions of pair-wise angles between the nearest Rubisco neighbours (from $n$ = 198,692, $n$ of tomograms = 26, particles with distance smaller than 10.5 nm were excluded). The position and orientation of each Rubisco particle were determined with C1 symmetry (i.e. no D4 symmetry was applied). **c** Mapping-back of Rubisco by pair-wise distances of nearest neighbours, Rubisco is coloured in various shades of green accordingly. **d** Mapping-back of Rubisco by pair-wise angles of nearest neighbours. Rubisco is coloured in various shades of green accordingly. The mapping back is exemplified in the pyrenoid of tomogram in Fig. 1b, using a tomographic slice of 50 nm. **e** Local clustering of Rubisco with pair-wise distances ranging from 135 to 215 Å, pair-wise angles from 10° to 30°, curvature from 70° to 110°, and the minimum number of cluster formation equal to 3. Six examples are illustrated and **vi** shows a spiral arrangement of Rubiscos. Mapping back is illustrated using the D4 symmetric class 12 map for visualization purposes.

accommodation of the reaction intermediates, and substrate release[10,16,19]. Additionally, conserved residues at the active site, such as the glycines and valine that maintain the hinge motion of loop 6, are linked to the underlying flexible mechanism as well as pyrenoid Rubisco's kinetic traits[10,46]. Although the different classes of Rubisco do not exhibit specific compartmentalization within the pyrenoid, nor do they originate from a particular cell (lamellae), they are distributed around the pyrenoid tubules at varying enrichment levels and distances from one another (Fig. 6a). This spatial arrangement implies a potential relationship between the conformational states and functional roles of pyrenoid Rubiscos. Such findings underscore the need for future high-resolution in-cell studies to elucidate the dynamic behavior of these enzymes within the pyrenoid.

We observe evidence of Rubisco binding proteins primarily at two interfaces: adjacent to the small subunits and the at interface between large subunit dimers. The small subunit binding site has been shown to be the interface for the EPYC1 linker protein or other pyrenoid proteins which share a similar binding motif[12,47]. The large subunit dimer interface has been shown to be the binding site of CsoS2 and CcmM linker proteins in α- or β-carboxysome Rubisco, respectively, as well as for the carbonic anhydrase enzyme, CsoSCA, from α-carboxysomes. Interestingly, there is a good match between the binding location of these carboxysome proteins and the extra densities in our 3D classes (Supplementary Fig. 10). It is yet to be determined which protein(s) bind to the pyrenoid Rubisco at the large subunit dimer interface and what their functions may be. Interestingly, the density for EPYC1 is particularly evident in the best-resolved asymmetric map, and is averaged out in the D4 symmetric maps, suggesting partial occupancy on the small Rubisco subunits. In contrast, the density associated with the large subunit dimer interface is evident in the D4 symmetrized maps and appears at a lower resolution. This indicates that a subset of the Rubisco population exhibits a high occupancy of these binding proteins. A third interface may be located next to the active site, as evident from the subtraction of class 12 from classes 7, 10, and 13 (Fig. 3b-d blue). This potential binding interface may be attributed to Rubisco activase, as observed in the cryo-EM structure of the complex between Rubisco and the cyanobacteria "Rubisco activase-like" protein[40]. Alternatively, this may be the site for the interaction of neighboring Rubisco complexes within the pyrenoid.

The spatial organization of Rubisco within the pyrenoid of *Chlamydomonas* appears largely stochastic, showing a similar distribution pattern as reported previously[36], although our analysis on the pair-wise distance between classes shows potential specific spatial arrangement for Rubiscos of different functional groups. This difference to the ordered packing seen in carboxysomes may be attributed to the nature of the linker proteins that facilitate Rubisco condensation. In the pyrenoid, EPYC1, a multivalent disordered protein, forms a dynamic network with Rubisco through LLPS[12]. This mechanism, while efficient in concentrating Rubisco dynamically and in forming of local clusters, likely lacks the rigid scaffolding required for overall precise geometric organization, leading to the observed stochastic distribution. In contrast, α-carboxysomes and β-carboxysomes employ CsoS2[31,32] and CcmM[13,33,48], respectively, which contain well-defined repeat domains that directly interact with Rubisco and the shell, imposing structural constraints that enforce a more ordered lattice-like arrangement[14,32,49]. These distinctions suggest that while the pyrenoid's phase-separated

architecture may provide flexibility in adapting to environmental changes, carboxysomes have evolved a more rigid assembly mechanism optimized for stability and efficient $CO_2$ fixation[50,51]. Future work dissecting the molecular interactions within these compartments may reveal how linker proteins influence Rubisco's distribution and catalytic efficiency in diverse $CO_2$-concentrating mechanisms.

In summary, this study offers direct observations of the structure and function of Rubisco within the native and intact pyrenoid, deepening our understanding of the carbon fixation mechanism in algae. Additionally, the study proposes a framework for further investigations on native Rubiscos at higher resolutions and in different conditions.

## Methods

### Strains and cell culture
*C. reinhardtii* wild-type strain CC-1690 was kindly given by Prof. Dr. Luning Liu at the University of Liverpool. Cells were maintained in Tris-acetate-phosphate (TAP) medium (Thermo-Fisher Scientific) under a normal circadian cycle (12-hour light and 12-hour dark) with the light intensity at 12,000 lux at room temperature on a shaker at 150 rpm.

### Plunge-freezing vitrification
Cells were harvested at 6 hours after entering the light cycle. An aliquot of 3.5 μl *C. reinhardtii* cell suspension at $1.5 \times 10^6$ cells/ml was applied to a glow-discharged holey carbon-coated copper grid (R 2/1, 200 mesh) (Quantifoil) and blotted from the back of the grid for 9 s using a Leica GP2 plunger (Leica Microsystems), followed by the plunge freezing in liquid ethane.

### Cryo-FIB milling
Lamellae were prepared using an Aquilos 2 cryo-FIB/SEM (Thermo Fisher Scientific) located at the electron Bio-Imaging Centre (eBIC), UK or using an Arctis plasma cryo-FIB/SEM (Thermo Fisher Scientific) located at the Rosalind Franklin Institute, UK (Supplementary Table 1).

Lamellae preparation on Aquilos 2: Thinning was performed on a rotatable cryo-stage maintained at −191 °C via an open nitrogen circuit. Prior to milling, grids were mounted onto a shuttle, transferred to the cryo-stage, and coated with a trimethyl (methylcyclopentadienyl) platinum (IV) (organometallic platinum) layer using the GIS system (Thermo Fisher Scientific) for 30 seconds. Cells located near the centers of grid squares were selected for thinning. The process was carried out in a stepwise manner using the automated milling software Auto-TEM (Thermo Fisher Scientific), with gallium beam currents decreasing incrementally from 0.5 nA to 30 pA at 30 kV. The final thickness of the lamellae was set to 120 nm.

Lamellae preparation on Arctis: Autogrid clipped TEM grids were loaded into the chamber via the robotic delivery device (Autoloader) in a dual-beam plasma focused ion beam scanning electron microscope with a redesigned sample stage, which is a prototype for the commercially available Arctis microscope (Thermo-Fisher Scientific). The ion species used for milling was Argon at 30 kV. Both ion and SEM columns are aligned, and the measured currents are within 20% of their target. SEM imaging was done on the grids to ensure suitability for lamella preparation. Once the sample was selected, a conductive layer has been deposited on the sample by milling a platinum target (16 kV 1.4 μA). Then a protective layer of organo-platinum was deposited

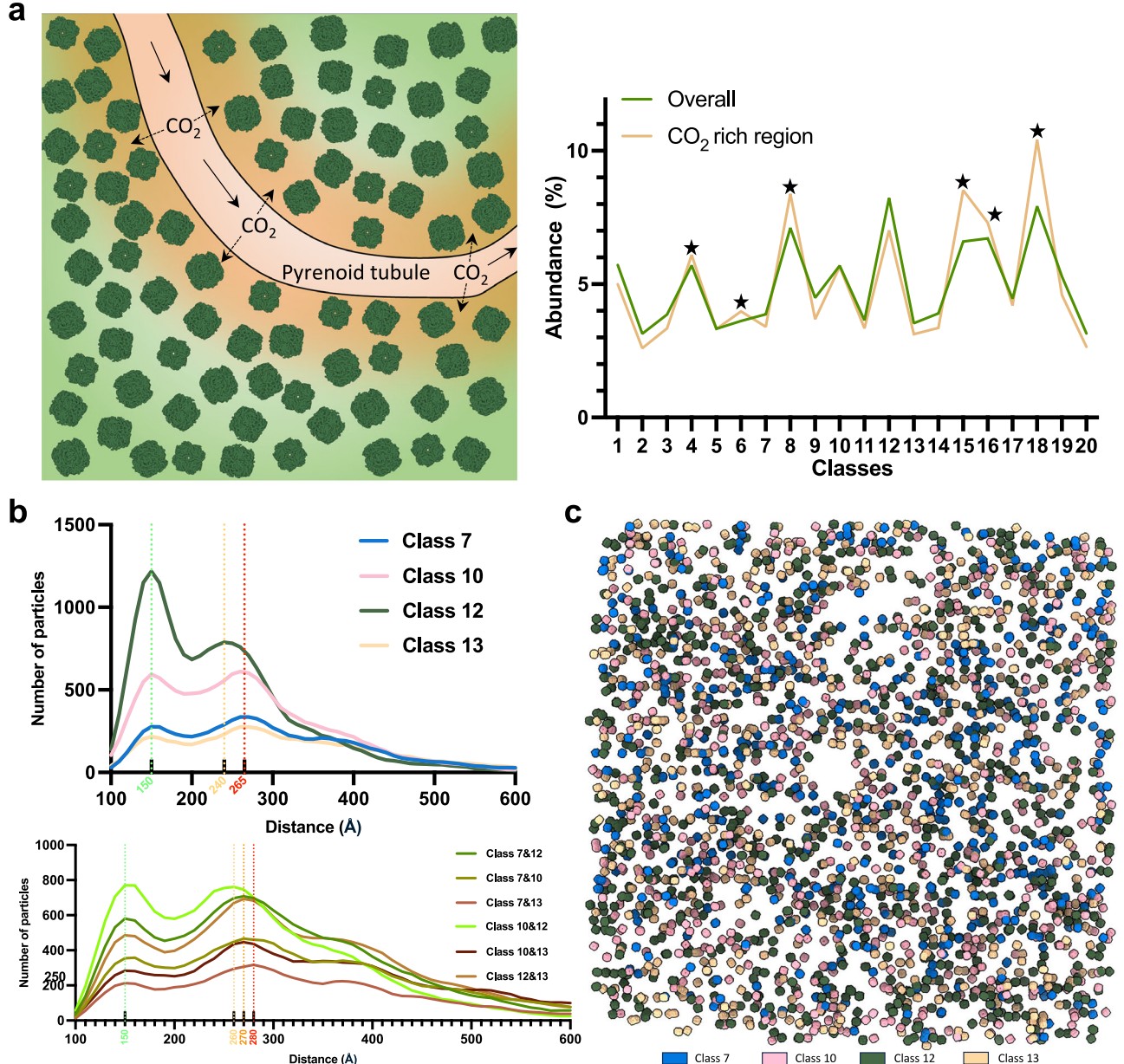

**Fig. 6 | Analysis of Rubisco distribution based on classes. a** Distribution of Rubisco particles in $CO_2$ rich regions. On the left, a schematic illustration depicts $CO_2$ diffusion from the pyrenoid tubules into surrounding areas. Regions within 200 nm of the tubule surface are regarded as $CO_2$-rich. On the right, a line plot shows the abundance of Rubisco particles across different classes, both overall and within $CO_2$-rich regions. Stars indicate classes that are enriched in $CO_2$-rich regions (Two-sided Chi-square test for all classes, $p < 0.0001$). **b** Distribution of pair-wise distances of the nearest neighbours in (upper) and across (lower) the best-resolved four classes. The distribution peaks are indicated by dashed lines (*n* of class 7 = 8339, *n* of class 10 = 12,267, *n* of class 12 = 17,713, *n* of class 13 = 7611, *n* of tomograms = 26). **c**, Mapping-back of Rubisco by classes, with four major classes included and coloured accordingly in the tomogram shown in Fig. 1b.

using a gas injection system. Finally, another conductive layer of platinum was deposited. Lamella sites were identified and queued in the AutoTEM software (ThermoFisher Scientific) before unattended milling using the following sequence: i) stress relief trenches (0.74 nA); ii) three steps coarse milling; (2, 0.74 and 0.2 nA) iii) two steps polishing (60 pA and 20 pA).

### Cryo-electron tomography data collection

Tilt series were acquired using two Titan Krios G4 TEMs (Thermo-Fisher Scientific) with similar setups, one at eBIC and one at RFI (Supplementary Table 2). Both microscopes were operated at 300 kV with fringe-free illumination. Imaging was done on a Falcon 4i direct detector installed behind a Selectris X energy filter (Thermo-Fisher Scientific), using a slit of 10 eV. All tilt series were recorded at a nominal magnification of 64,000×, corresponding to physical pixel sizes of 1.97 Å (eBIC) or 1.978 (RFI), using the dose-symmetric scheme starting from the lamella pre-tilt of −12° or 12° (dependent on grid orientation) generated from cryo-FIB milling, 2° increments and tilt span of 54°. Tilt series were acquired using an automated low-dose procedure implemented in Tomography 5 software (Thermo-Fisher Scientific). In some positions multiple tilt series were acquired using image-beam shift, however in most lamellae only one tilt series could be fitted in the pyrenoid area. The nominal defocus range was 2.5 to 4.5 μm, and the total dose was ~120 e⁻/Å². Each tilt series was fractionated into 6 movie frames.

### Subtomogram averaging

Movie frames were motion-corrected using MotionCor2[52], followed by CTF estimation within individual tilts using CTFFIND4[53] and tilt-series

alignment and reconstruction in bin 2 (3.94 Å/pixel) using AreTomo[54] and IMOD[55]. In total 26 tilt-series were selected for analysis based on the quality of the reconstruction and appearance of extensive Rubisco clusters. Alignment quality was mainly assessed by mean residual error, selecting tomograms that showed values smaller than 1 nm for successive data analysis. Particle picking was done using the deep learning based crYOLO software (v1.9.3)[56]. The network was trained by manually annotating 12 sections from 3 different tomograms reconstructed in bin 2. Training was done using "crYOLO" architecture, low pass filtering frequency of 0.1 and 38-pixel boxes. CrYOLO provides a confidence value for each particle following prediction, and the user can adjust the confidence threshold for optimizing the picking results. We found that the default prediction parameters gave excellent results (confidence threshold 0.3. Tomography picking mode: tracing search range 25% of box, tracing memory 0 and tracing min length 5), resulting in 515,948 subtomograms from all tomograms (Supplementary Fig. 1). Tilt-series, along with corresponding alignment files, CTF parameters, and particle coordinates, were imported into RELION4[57] for running the STA pipeline. Subtomograms (RELION PseudoSubTomograms) were initially processed in bin 4 (7.88 Å/pixel) and $32^3$ box size. "Bad" subtomograms were discarded by two rounds of 3D classification into 10 classes, without applying symmetry. "Good" 3D classes were selected based on the overall appearance of Rubisco features, the map's resolution, and accuracy in the angular assignment. For example, 3D classes 4, 6, 8, 15 and 18 in Supplementary Fig. 2c, all feature different levels of distortions compared to the rest of the classes and were therefore considered as "bad". We found that accuracy in the angular assignment (_rlnAccuracyRotations) to be a particularly indicative parameter for assessing the quality of 3D classes in 3D classification and 3D auto-refine jobs. The SPA map of *C. reinhardtii* Rubisco filtered to 60 Å was used as an initial reference (EMD-22401[12]), and a mask of 180 Å diameter was applied during the process. The resulting 215,272 subtomograms were 3D classified into 20 classes with a mask of 160 Å diameter applied (Supplementary Fig. 2c). Refinement of all good classes was then performed at bin 2 (3.94 Å/pixel), box size $100^3$, with C1 or D4 symmetry applied, and mask diameters of 160 or 150 Å, respectively (Supplementary Fig. 2d, e, Supplementary Table 3). The best-resolved D4-symmetrized map (class 12, 17,713 subtomograms) was further refined at bin 1 (1.97 Å/pixel) and $256^3$ box size. CTF refinement ($512^3$ box size for CTF estimation) followed by frame alignment and reconstruction with an applied solvent mask resulted in a final resolution of 8.1 Å (Fig. 2a, Supplementary Fig. 2f and 3a,b Supplementary Table 4).

Difference maps between classes (Fig. 3b-g) were prepared as follows: refined maps of classes 7 and 12 were low pass filtered to 13.6 Å to match the lower resolution of classes 10 and 13. Confidence maps were calculated with a noise box size of 27 pixels[38], binarized at 1% false discovery rate threshold[38] in ChimeraX[58], and subtracted from each other. The resulting difference density was Gaussian filtered (SD = 4) and displayed at a contour level of 10 SDs with "hide dust" set to 15 Å. The difference map in Fig. 4b was prepared in the same way, except that the low pass filtered, D4 symmetric class 12 map was subtracted from the corresponding C1 map.

### Real-space refinement of Rubisco coordinates within the cryo-EM map

A specialized real-space refinement protocol was developed for the present work (Supplementary Fig. 11). The protocol is detailed in the Supplementary Information and is summarized here. First, rigid-body docking was performed to embed the large subunits (PDB: 1GK8[15]) and small subunits (PDB: 1EJ7[20]) of the Rubisco structure into the best-resolved D4-symmetrized class 12 map. Mutations introduced during crystallization were reverted back to the wild-type sequence (UniProt ID P00877 for the large subunit and P00873 for the small subunit) using ChimeraX[58]. Flexible regions, namely loop 6, the N- and C-termini

of the large subunit were generated de novo using Rosetta version 2024.09[59–62]. The C-terminus of the large subunit was restrained by maintaining the distances between D473-R134 and D473-H310, to guide the model into the cryo-EM density, as the interactions were observed in previous studies[20]. The model was further refined iteratively by applying a local rebuilding procedure using CartesianSampler mover in Rosetta. Protonation states of titratable groups were determined by using PD2PQR[63], then ions and water molecules were added using VMD[64]. Molecular dynamics flexible fitting (MDFF) simulations[65,66] were performed as described in Perilla et al.[67], while the backbone heavy atoms were coupled to the Cryo-EM density map using a grid-based biasing-potential[68].

The MDFF-derived model revealed multiple solutions for the N- and C-termini coordinates of the large subunits, while in none of the chains both termini were fitted in the EM density simultaneously (Supplementary Fig. 12). Thus, a hybrid large subunit was created by combining residues 1–103, taken from the best-fitting N-terminus, and residues 104–475, taken from the best-fitting C-terminus chain. The complete Rubisco structure built from the hybrid large subunit model was subjected to further relaxation to remove possible steric clashes using NAMD3.0.1.

The minimized-hybrid model was further refined iteratively by using an unsupervised[69] and supervised protein refinement tool, applying FastRelax[70–72] protocol in Rosetta on selected regions. The unsupervised refinement tool identified regions with low molProbity score[73,74], while the supervised refinement tool selected regions having poor local-cross correlation (Supplementary Fig. 13). To improve the quality of the beta sheets, the beta strands coordinates on the large subunits were replaced by the coordinates of the corresponding residues in PDB 1GK8 (Supplementary Fig. 14). Another round of symmetrical MDFF simulation were performed on unprotonated Rubisco structure. Lastly the model was subjected to minimization using NAMD3.0.1 without coupling the protein to the cryo-EM density. The resulting model was in good agreement with the density, as evidenced by the model to density FSC (Supplementary Fig. 15d, Supplementary Table 4), and the local cross-correlation (Supplementary Fig. 15e, f). Segment-based Mander's overlap coefficient (SMOC)[75] and residue-based Q-score[76] are presented in Supplementary Fig. 16 and 17. SMOC between the model and cryo-EM density was calculated as $0.87 \pm 0.04$ and $0.90 \pm 0.02$ for the large and small subunits, respectively. Global Q-score was calculated as 0.28, in agreement a resolution of 8.1 Å. Q-scores in the active site are in good agreement with values for the resolution of the density indicating a good match between residues and the map (Supplementary Fig. 17).

### Segmentation

To enhance the segmentation, reconstructed tomograms at bin 4 were corrected for missing wedge and denoised using IsoNet[77] version 0.2, applying 35 iterations with sequential noise cut-off levels of 0.05, 0.1, 0.15, 0.2, 0.25 at iterations 10, 15, 20, 25, 30, respectively. Thylakoids and pyrenoid tubules were initially segmented using MemBrain-seg[78], and then imported into ChimeraX[58] for cleaning and polishing. Rubisco particles were mapped back into the tomographic space based on their refined positions and orientations. The Rubisco structure in the segmentation was adopted from the class 12 map in this study.

### Measurements of pair-wise distance and angle

Particles included in this analysis ($n = 215,272$) were originally picked using crYOLO, followed by the removal of "junk" using iterative classification without imposing symmetry in RELION4, as described above (Supplementary Table 3, Supplementary Fig. 2). Distance and angular measurements were performed using the C1-refined reconstructions (without applying D4 symmetry), while the D4 symmetric class 12 map was used for the placement in map back for display purposes only. The distance between adjacent Rubisco particles was calculated according

to the coordinates of their centers after the refinement. Paired particles with a center-to-center distance shorter than 105 Å were regarded as duplicates and removed. To analyze the distribution of Rubisco in $CO_2$-rich regions, the pyrenoid tubules were segmented and divided into separate sampling points using the open-source software PyVista (https://docs.pyvista.org/). The distance between the sampling points and Rubisco particles was then calculated, and particles with a distance shorter than 100 nm from the nearest sampling point were retained. For the measurements of distance and angle between classes, particles were first divided into separate datasets based on their class number. The distribution and local clustering analysis of Rubisco were performed using the software MagpiEM (https://github.com/fnight128/MagpiEM). The statistical analyses were performed in Prism 10. The clustering analysis was conducted in MagpiEM by restricting the pair-wise distance to 105–215 Å, pair-wise angle to 10–30°, curvature to 70–110°, and the minimum number for cluster formation to 3.

### Reporting summary

Further information on research design is available in the Nature Portfolio Reporting Summary linked to this article.

## Data availability

Raw tilt series have been deposited in EMPIAR (EMPIAR-12515). The best-resolved (class 12) subtomogram averaging map and corresponding model have been deposited in the Electron Microscopy Data Bank (EMDB) and Protein Data Bank (PDB), with accession codes EMD-52438 and 9HVM, respectively. Subtomogram averaging maps from D4-symmetric classes 7, 10 and 13, and asymmetric class 12 map, have been deposited as additional maps under the same EMDB entry. Source data are provided with this paper.

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

## Acknowledgements

We thank Daniel K. Clare for support and David Farmer for help with scripting. We thank Diamond Light Source for access and support of the cryo-EM facilities at the UK National Electron Bio-Imaging Centre (eBIC) (proposal NT29812 and NR21005). The work was supported in part by a grant from the Estate of Louise Yasgour, the UK Wellcome Trust Investigator Award (206422/Z/17/Z), the UK Wellcome Discovery Award (311427/Z/24/Z), ERC AdG grant (101021133), and National Institutes of Health grants (U54 AI170791, P50AI150481 and R21AI184080). This work used Stampede3 at TACC through allocation MCB-170096 from the Advanced Cyberinfrastructure Coordination Ecosystem: Services & Support (ACCESS) program, which is supported by National Science Foundation awards (2138259, 2138286, 2138307, 2137603, and 2138296). Arctis development is supported by the Wellcome Trust through the Electrifying Life Science grant (220526/Z/20/Z to J.H.N.). The Rosalind Franklin Institute is funded by the UK Research and Innovation, Engineering and Physical Sciences Research Council.

## Author contributions

N.E., Z.H. and P.Z. conceptualized and designed the study. Z.H., M.D. and N.E. prepared the cryo-EM samples and collected cryo-ET data. N.E. and Z.H. analyzed the data. A.R. and J.R.P. performed real-space coordinates refinement within the cryo-EM map. N.E., Z.H. and P.Z. wrote the manuscript with contributions from all authors.

## Competing interests

The authors declare no competing interests.
