## [Transparent Peer Review file · Nature Communications]

In-cell Structure and Variability of Pyrenoid Rubisco

Corresponding Author: Professor Peijun Zhang

Version 0:

Reviewer comments:

Reviewer #1

(Remarks to the Author)

In this manuscript, the authors characterized the cellular architecture of pyrenoid inside *Chlamydomonas* cells, a liquid-like compartment important for the efficiency of CO₂ fixation and for bioengineering applications. The authors performed cryo-electron tomography (ET) of cryo-focused ion beam milled *Chlamydomonas* cells, subtomogram averaging (STA) of Rubisco particles, molecular dynamics flexible fitting (MDFF) for the resulted EM map, as well as analysis of the Rubisco distribution within the cellular pyrenoid compartment, which is state of the art. Although the Rubisco complex appears as a special case for STA as they are abundant in the pyrenoid, I still feel that the methodology provided in this manuscript will advance the field by providing strategies for analyzing structures of relatively small cellular complexes like Rubisco (~13 nm), their structural variations, and their distribution profiles directly inside the cellular context.

I think there are a few points that could be further clarified:

1. In the manuscript, the authors selected 26 tilt series out of 66 (37+29) lamellae that are collected, considering the tomogram quality. It seems like the number of usable tilt series are considerably low, given the fact that normally at least about 5 tilt series per lamella can be collected with the settings described in the method section. I wonder if there are specific technical reasons that limit obtaining more tilt series of the pyrenoid regions from these lamellae. Or, is this due to a strict selection criterion of tomograms? If so, could the authors please provide supplementary figures or description in the text for the reasoning? This is of importance as such factors often relate to the final resolution of the structures obtainable by STA.
2. In the STA pipeline, the authors classified the subtomogram particles into 20 initial classes. Out of these, the best class (of 17,713 particles with D4 symmetry applied) gave a resolution of 8 Å, with a pixel size of 1.97 to 1.98 Å used for data collection. Is there a possibility that some good particles were misclassified into the classes that they should not belong to? Could this lead to a relatively small percentage of particles contributing to the final high-resolution information of the EM map? If the authors have tried different number of classes to start with and found a number of 20 useful for this case, please include it in the description for the readers to understand the rationale.
3. Regarding the stochastic distribution of Rubisco particles, did the authors profile the pair-wise distances and angles of different classes relative to each other, in order to reveal possible coordination of conformations of Rubisco within the pyrenoid? Also, some more discussion on the difference of Rubisco distribution within the pyrenoid from that in the α - and β -carboxysomes and possible factors contributing to this would be useful for understanding the general mechanism.

Minor points:

1. Line 56-57, "The active site is located at the interface between the C-terminal of one large subunit and the N-terminal of an adjacent one, such that two large subunits form a functional dimer harboring two active sites." It would be useful to illustrate this with the resolved model/map in the figures.
2. Supplementary Fig. 9 (e-g), for superposition, could the author please use different colors for two structures? I also wonder if the pictures in (a) and (b) are consistent with the legend, as the N-ter model looks slightly better in (b) than that in (a), different from the description in the legend.

3. Legend of Supplementary Fig. 12, “for (e) large and small subunits.”, (f) is missing in the text.

Reviewer #2

(Remarks to the Author)

Summary

This manuscript presents the structural analysis of RuBisCOs within the pyrenoid of *Chlamydomonas reinhardtii* using cryo-electron tomography (cryo-ET) in combination with cryo-focused ion beam (cryo-FIB) milling, subtomogram averaging (STA), and molecular dynamics flexible fitting (MDFF). The authors revealed significant structural heterogeneity of RuBisCO complexes, including a closed (activated) conformation, local conformational changes at the active site, and interactions with binding proteins. These findings highlight the dynamic nature of RuBisCO in its native cellular environment. The study also explores the stochastic distribution of RuBisCO particles within the pyrenoid and the potential roles of binding proteins, particularly EPYC1. Overall, these findings make an important contribution to our understanding of RuBisCO in its native context within the pyrenoid. However, the following revisions would strengthen the manuscript.

Major comments:

1. The observation of structural heterogeneity of RuBisCO complexes within the pyrenoid is one of the most compelling results of this study. The authors do an excellent job of identifying different conformational states of RuBisCO (closed and open conformations) and providing insights into how this heterogeneity impacts RuBisCO's function. However, the manuscript could do more to quantify the extent of this heterogeneity and its implications for RuBisCO's catalytic activity. For instance, how do the authors identify different conformational states of RuBisCO at such rather low resolutions? Could you provide a quantitative summary of the relative abundances of different RuBisCO conformational states observed in your study? How does the distribution of these conformational states correlate with RuBisCO's enzymatic function within the pyrenoid? A more direct link to functional implications could strengthen this section.
2. The identification of potential binding proteins, specifically EPYC1, interacting with RuBisCO is another major contribution of this paper. While the interaction between RuBisCO and EPYC1 is compelling and well-supported by previous studies, the manuscript introduces an additional potential binding site at the interface between RuBisCO large subunit dimers. However, identification and functional validation of these binding proteins remains speculative. If this interaction involves an unknown protein, the authors could speculate about its identity or functionality by conducting a sequence homology search using known RuBisCO-binding partners. Alternatively, biochemical assays such as co-immunoprecipitation (Co-IP) or cross-linking mass spectrometry could be employed to identify the bona fide binding partners of RuBisCO in *Chlamydomonas reinhardtii*.
3. The results showed that RuBisCO particles are distributed in a stochastic manner within the pyrenoid. It is interesting as it contrasts to the rigid, ordered packaging in carboxysomes. While the random distribution of RuBisCO particles is a significant and novel observation, the authors briefly mention the existence of local clusters of RuBisCO particles, but the functional significance of these clusters is not fully explored. The authors could enhance this section by conducting statistical analysis to further investigate the characteristics of these local clusters. For example, do these clusters somewhat resemble the spiral, fiber, or filamentous arrangements seen in carboxysomes? Whether the RuBisCO particles directly interact with each other within the clusters or could these interactions be mediated by EPYC1 or other binding proteins? Would you propose that there is a correlation between RuBisCO clustering and areas of higher CO₂ fixation activity in the pyrenoid?
4. The manuscript provides structural insights into the conformational changes around the active site of RuBisCO, including movements in the key loops (e.g., loop 6 and the N- and C-termini) that regulate substrate binding and catalysis. These findings are important for understanding RuBisCO's catalytic mechanism. However, given the relatively low resolution of the structural data, it is important to consider how confidently the authors can assign specific conformational states, particularly in the key loop regions of RuBisCO. More critical evaluations on the quality of the cryo-EM maps and fitting of the MDFF models should be provided. Furthermore, to enhance the novelty and impact of the manuscript, the authors could discuss how the conformational flexibility of these loops could affect the catalytic efficiency of RuBisCO under physiological conditions in response to varying CO₂ concentrations.

Minor comments:

1. The authors note the high level of structural heterogeneity in the dataset, which led to a highest resolution of 8.1 Å. The manuscript suggests that heterogeneity, including conformational variability and binding protein interactions, is a significant factor in reducing signal-to-noise ratio (SNR) and thus the achievable resolution. During data processing, while the authors used iterative 3D classification, more advanced techniques could be applied to handle heterogeneity more effectively. For example, multi-class averaging or focused refinement could help extract more accurate structural details from heterogeneous populations.
2. In the Methods section, the tilt series acquisition protocol is detailed and well-explained. However, the manuscript does not mention the rationale behind the tilt range (−12° to 12°) that. Tilt ranges can influence the final resolution, and a more explicit justification of these parameters would be helpful.
3. The methodology for particle picking and subtomogram averaging is well-described. However, the authors mention two rounds of 3D classification to discard “bad” subtomograms. The criteria for classifying “good” and “bad” subtomograms would benefit from further elaboration. How were “bad” subtomograms identified specifically (e.g., based on image quality, presence of distortions, lack of RuBisCO features)?
4. A deep learning-based method crYOLO was used for particle picking. It would be beneficial to provide more details on this method and how the network was trained. This will give the reader a better understanding of the network's performance and its ability to accurately detect RuBisCO particles.
5. Lines 81-82. The recently published paper describing the intact structure of α-carboxysome should be cited here (Nature Plants. 2024 Apr; 10(4):661-672).

6. Please provide all relevant cryo-EM maps for the reviewers to evaluate their qualities and the reliability of fitting of the models into the maps.

7. Figure 5 should be revised for more clarity. Currently, it does not obviously illustrate the arrangement of RuBisCO particles and their clustering in the pyrenoid.

Reviewer #3

(Remarks to the Author)

Reviewer #4

(Remarks to the Author)

Elad et al., present a detailed structural analysis of in situ sub-tomogram averages of the crucial CO₂ fixation protein Rubisco in the model green alga *Chlamydomonas reinhardtii*. They achieved the highest resolution structure of in situ Rubisco at 8.1 Å, which allowed previously published Rubisco coordinates to be refined within their map using Molecular Dynamics Flexible Fitting. The results support that the best resolved classes closely resemble the published open and closed Rubisco conformations, and demonstrate that Rubisco does not show large whole domain movements. Moreover, Rubisco is shown to display a large heterogeneity, which aside from being a result of the aforementioned open and closed conformation stems also from the presence of additional densities at the large subunit dimer-dimer interfaces and could potentially represent densities derived from yet unknown binding partners. The authors also investigate the spatial distribution of the different Rubisco states, and they do not find any compartmentalisation of different Rubisco classes. They are also able to corroborate previous results on Rubisco's nearest neighbour distances.

Whilst the findings are interesting, they predominantly are descriptive and give little new biological insight into pyrenoid function. Much of the data supports previous cryo-ET data performed on the *Chlamydomonas* pyrenoid. Including the detailed description of pyrenoid architecture provided in Engel et al. *eLife* (2015) and that Rubisco has a "liquid-like" organisation in the pyrenoid demonstrated by Freeman Rosenzweig et al. *Cell* (2017). Whilst identifying open and closed Rubisco states and additional densities on Rubisco provides advances there is a lack of biological insight or further experimental investigation. For example, it could have been interesting to look at Rubisco states between different relevant environmental conditions such as photoautotrophic growth at high and low CO₂. Whereas all the images are taken on mixotrophic grown cells where there is an unclear relationship between photosynthetic carbon fixation and acetate usage. The manuscript also fails to look at broader pyrenoid architecture, such as how the Rubisco matrix interfaces with the pyrenoid tubules or the starch sheath. The described work within a manuscript that looked at wider architecture would give better insights, although performing this on photoautotrophic grown cells would give more biologically relevant insights when the pyrenoid is fully operational within the CCM.

Reviewer knowledge gaps: The reviewers would like to indicate that assessing the MDFF modelling is outside the scope of their expertise.

Major comments:

1. The two paragraphs exploring the local variations between the four analysed classes are written very confusingly. For instance, "Interestingly, significant differences (density gain) occur between class 12 and all other classes at the active site (Fig. 3b-d blue)." This initially suggested that it is class 12 that has additional density compared to the other classes, when figure 3 shows the reverse to be true. I had the same issue with "As observed for the active site, class 12 shows major differences (density loss) at this interface compared to the other 3 classes (Fig. 3b-d, green)." It would be good to re-write this section to make it much clearer where the gain and loss is and relative to which classes.
2. Authors point out that the additional densities around the active site in classes 7, 10 and 13 most likely correspond to the mobility of the loop 6, N- and C-termini and loop 64-68 and relate to classes 7, 10 and 13 being in the open conformation, while class 12 is in the closed confirmation. This area is also where Rubisco activase is binding and performing its activity (Flecken et al., 2020). Could the authors comment whether any of the detected additional densities around the catalytic site could correspond to Rubisco activase?
3. Authors show the presence of additional densities at the polar regions of C1 refined class 12 Rubisco, adjacent to the small subunit which they propose to be the EPYC1 protein bound at sub-stoichiometric concentration. Have the authors attempted symmetry expansion and local refinement to potentially increase the resolution of the region of interest?
4. The authors then take their particles from the four best resolved classes to assess their spatial organisation within the pyrenoid by mapping the pairwise distances and angles. In the text the authors state that the nearest neighbour analysis is of particles in class 12 only (line 246), but then in the figure the particle number is 215,272, meaning it contains the particles distributed in the 20 classes from the sub tomogram averaging. In this context it would be useful to understand how did the authors pick the "true" Rubiscos to be included in the analysis. Were these all crYOLLO picks? As we are (and potentially other readers) not familiar with this software it would be useful to explain whether it provides scoring of picked particles and what was the thought process behind including these particular picks.
5. The methods section indicates that the *Chlamydomonas* cells were grown mixotrophically in TAP media, similarly to the method used by Engel et al., and Freeman Rosenzweig et al. It appears that currently there is not much information on the impact of heterotrophic growth on photosynthesis under low CO₂, however isotope labelling has shown that even at 5%

CO₂ as much as 22% of carbon was derived from acetate (Heifetz et al., 2000). In this context it is hard to conclude how generalisable the findings on the pyrenoid organisation are. Could authors elaborate on why did they choose to grow their cells mixotrophically and how generalisable the results are to photosynthesis by providing a comparison to TP grown cells or comparing the distribution of Rubisco at the growth conditions using TAP and TP media with fluorescent microscopy. Moreover, the fraction of Rubisco in the pyrenoid in synchronised cultures varies depending on the time in the day-night cycle (Mitchell et al., 2014), so could the authors specify at what point in the dark-light cycle did they vitrify the cells.

Minor comments:

78: "The difference in binding sites on Rubisco by these linker proteins is believed to be related to the packaging of Rubisco." What is the support for this statement? Recent work on the *Chlorella* pyrenoid has shown that CsLinker (an EPYC1 analog) binds to the Rubisco large subunit (Barrett et al. 2024, Nature Plants).

79-81: "As the packaging of Rubisco affects the utilization and assimilation of CO₂, understanding the in-situ organization of Rubisco particles has garnered significant attention" - Could the authors provide a reference for how different packing affects the utilization of assimilation of CO₂? Do the authors mean different packing between carboxysomes, pyrenoids and plants not containing CCM, or the impact of different packings in different pyrenoids?

81-84: "Recent studies have demonstrated intriguing packaging patterns for Rubisco in α - and β -carboxysomes which are confined by shells^{15,33,34}. Whether similar or distinguished packaging patterns exist for Rubisco in the pyrenoid remains elusive."

Freeman Rosenzweig et al. 2017 demonstrate using cryo-ET that Rubisco packaging in the pyrenoid is liquid-like, with no clear organised packaging. The authors do discuss that their data agrees with this in the discussion, so it is a bit misleading to state it as remaining elusive.

98-99: "Moreover, the distribution of Rubisco in pyrenoid is revealed to be stochastic, different from that rigid packaging in the α - and β -carboxysomes." The *Chlamydomonas* pyrenoid has been shown to display liquid-liquid phase separation. Is the finding of the stochastic distribution of Rubisco and its different classes something the authors consider novel and not expected given the LLPS?

119: The demonstration of heterogeneity of Rubisco within the pyrenoid is very interesting. Could the authors show the angular distribution of the four best classes to demonstrate that the difference in densities is not due to missing views?

111-112: "To resolve the structure of Rubisco, deep learning-based particle picking was performed, and most discernible Rubisco particles were correctly picked (Fig. 1c)." -What stage of processing are the particles from figure 1c derived from? Are these all crYOLo picks or are there the result of classification and "junk" removal?

133: It would be helpful to mention that figure 2a is a result of MDFF docking when it is brought up first time (so line 133 not 137).

166: typo - Asp437 should be Asp473 (as per cited references).

186: Can the authors include how prevalent each of the analysed classes is in the particle population.

214-216: The authors also mention the densities at the large subunit dimer interfaces to be reminiscent of the areas of carboxysome matrix "and likely indicates the presence of Rubisco binding proteins within the pyrenoid." The phrasing of "likely indicates the presence of Rubisco binding proteins within the pyrenoid" suggest as if the presence of Rubisco binding proteins was a novel discovery, despite the fact that we know of many proteins containing EPYC-like Rubisco binding motifs and of Rubisco-activase. It would be more appropriate to say it indicated the presence of yet-unidentified proteins likely possessing novel mode of Rubisco binding for *Chlamydomonas*, but identified in carboxysomes.

234: "The potential EPYC1-associated" – typo

249: "but slightly larger than the distance in β -carboxysomes (122 nm)" – typo, missing comma.

252: Can the authors specify what angle and distance constrains they used for the clustering?

278: "Consequently, we find no evidence for synchronization or compartmentalization of Rubisco dynamics" – I understand the lack of evidence of compartmentalisation, but I wonder if synchronisation is the right word here? Since the cells were grown in dark-light cycles, then in principle they should be synchronised in regards to their cell cycle. Therefore, if there is any temporal dependency of the Rubisco dynamics it could be that the differences would not be detectable? Different grids were plunged at different time during the day-night cycle, which is something that should be specified.

The light intensity is given as lux, which does not relate to the Photosynthetic Active Radiation. Assuming that the lights used for growth were fluorescent lamps, then a rough calculation gives us 12,000 lux to be $\sim 162 \mu\text{mol m}^{-2} \text{s}^{-1}$ in photosynthetic photon flux density.

Figures:

Figure 1: Nomenclature doesn't align with current field: Pyrenoid tubules are tubular thylakoids running through the pyrenoid

matrix and minitubules are the small tubules found within the pyrenoid tubules that provide connections between the matrix and stroma.

Supplementary figure 2. Please label the FSC curves.

Figure 3: Add a not-lowpass filtered maps of the four best classes to the figure (currently only in the supplement).

References:

Most of the references in this paper are relevant with a few mistakes (see comments below), however in a few instances in the introduction the authors refer to multiple review papers instead of primary sources. Moreover, some of the introduction sentences contain multiple separate statements, that need references, but the authors often place them at the end of the sentence. This makes it difficult to locate which publication contains the relevant information, especially if they are broad review papers. Some examples include lines 46, 48, 66, 72.

42: “One-third of the global CO₂ is arguably fixed in algae by ribulose-1,5-bisphosphate carboxylase oxygenase, commonly known as Rubisco” – the paper says that 1/3 of photosynthesis is fixed by pyrenoids. Not all algae have pyrenoids, and some land plants such as hornworts do have pyrenoids. Moreover, the review is not the primary source of this information, but it is Mackinder et al. 2016 (referenced by ref5).

55: “Numerous structures of purified Rubisco have revealed the catalytic cycle in detail” - here a reference to a review discussing these Rubisco structures and the catalytic cycle would be appropriate

66: Reference 22 does not refer to Chlamydomonas.

67: “The pyrenoid is primarily found in many eukaryotic algae and plays a crucial role in enhancing the efficiency of carbon fixation, a process believed to have been driven by the gradual decrease of atmospheric CO₂ over billions of years until the recent industrial revolution by humans^{21–25}.” (Elad et al., p. 2) – This is an example of the multiple statements that would be appropriate to reference separately, i.e. reference to where the pyrenoid is found and then a reference to the pyrenoid evolution.

78-79 – “The difference in binding sites on Rubisco by these linker proteins is believed to be related to the packaging of Rubisco.” It would be good to add relevant references to this sentence.

82: the following reference also appears to be also relevant: <https://doi.org/10.1016/j.str.2024.05.007>

399 – reference 50 refers to Rosetta and is inconsistent with the reference cited in line 198 (ref 37).

References:

Flecken, M., Wang, H., Popilka, L., Hartl, F.U., Bracher, A., Hayer-Hartl, M., 2020. Dual Functions of a Rubisco Activase in Metabolic Repair and Recruitment to Carboxysomes. *Cell* 183, 457-473.e20. <https://doi.org/10.1016/j.cell.2020.09.010>
Heifetz, P.B., Förster, B., Osmond, C.B., Giles, L.J., Boynton, J.E., 2000. Effects of Acetate on Facultative Autotrophy in *Chlamydomonas reinhardtii* Assessed by Photosynthetic Measurements and Stable Isotope Analyses¹. *Plant Physiology* 122, 1439–1446. <https://doi.org/10.1104/pp.122.4.1439>
Mitchell, M.C., Meyer, M.T., Griffiths, H., 2014. Dynamics of Carbon-Concentrating Mechanism Induction and Protein Relocalization during the Dark-to-Light Transition in Synchronized *Chlamydomonas reinhardtii*. *Plant Physiology* 166, 1073–1082. <https://doi.org/10.1104/pp.114.246918>

Reviewer #5

(Remarks to the Author)

Version 1:

Reviewer comments:

Reviewer #1

(Remarks to the Author)

Elad et al aimed to reveal the native structural arrangement and dynamics of Rubisco complex within a model system *Chlamydomonas reinhardtii*. In the revised manuscript, the authors clarified a few major points regarding the sample preparation, classification procedures in data processing, and post-processing analysis of distribution of Rubisco

complexes. They mapped back the Rubisco into the tomograms and performed local clustering analysis for detailed illustration of potential higher-order assembly/coordination of Rubisco complexes. Also, the authors plotted the distribution of different classes of the complex in relation to the Pyrenoid tubule, as well as the relative distances/orientations of the best four classes based on the tomogram data, and concluded that the particles exhibit a predominantly stochastic distribution, with short-range local clustering of some particles in the pyrenoid. Regarding the binding proteins, the authors also performed further comparison and docking of previous models into their map and strengthened the point. Although the resolution of the maps remains insufficient to pinpoint the exact partner proteins, the study demonstrates the complexity of in-cell cryo-ET analysis, and future work with integrative strategies will hopefully help answer new questions brought out by the current manuscript.

I just have one remaining question that can be clarified for the pair-wise angle measurement. As the authors state the Class 12 model was used for the mapping-back analysis, please include in the method section whether or not the D4 symmetry has been included/considered during this analysis and if this could affect the angular distribution to some extent.

Reviewer #2

(Remarks to the Author)

In the revised manuscript, the authors have addressed most major concerns raised in the initial review. The manuscript is now scientifically rigorous, clearly written, and increases our understanding of the assembly pattern of Rubiscos in the pyrenoid. The cryo-ET combined with cryo-FIB milling is really a powerful technique for investigating the structural dynamics of pyrenoid Rubiscos in the native environment. This work also paves the way for studying the structural and functional correlations of the pyrenoid Rubiscos for enhanced CO₂ assimilation. Therefore I support publication of this manuscript in its current form.

Reviewer #3

(Remarks to the Author)

Reviewer #4

(Remarks to the Author)

We would like to thank the authors for thoughtfully addressing our comments. The new provided figure 6 with classes in relation to tubules is interesting and a good addition to the manuscript. The authors have addressed all of the statements we thought were problematic and added a stronger emphasis on the structural aspect of the work. We think it is reasonable for them to say that getting more tomograms would be out of the scope of this study.

There are still some minor referencing issues with the last comment addressed:

Initial comment and rebuttal: "399 – reference 50 refers to Rosetta and is inconsistent with the reference cited in line 198 (ref 37).

Response: We think that there is a misunderstanding here.

The first reference is for ChimeraX software: Goddard, T. D. et al. UCSF ChimeraX: Meeting modern challenges in visualization and analysis. *Protein Science* 27, 14–25 (2018).

The second reference relates to the "1% false discovery rate threshold" method: Beckers, M., Jakobi, A. J. & Sachse, C. Thresholding of cryo-EM density maps by false discovery rate control. *IUCrJ* 6, 18–33 (2019).

Rosetta software is referenced elsewhere."

We would like to clarify our initial comment but also highlight a referencing issue:

In line 466 (previously 399) the fragment "binarized at 1% false discovery rate threshold in ChimeraX (ref 50)" was initially a reference to Rohl, et al., 2004, which refers to Rosetta and not to the false discovery rate nor ChimeraX. However reference 50 is now: Zang, K., Wang, H., Hartl, F. U. & Hayer-Hartl, M. Scaffolding protein CcmM directs multiprotein phase separation in β -carboxysome biogenesis. *Nat Struct Mol Biol* 28, 909–922 (2021).

It appears that the references maybe out of sync and need to be checked.

Version 2:

Reviewer comments:

Reviewer #1

(Remarks to the Author)

I would like to thank the authors for clarifying the question. But I think the second part of the question has not been answered. As the authors pointed out, "the Distance and angular measurements were performed using the C1-refined reconstructions". However, the D4 symmetry was applied for most structural analyses of this manuscript and thus a majority of results here have been interpreted with the symmetry applied, implying that the D4-symmetry is biologically meaningful. Based on the definition of D4 symmetry operation, the maximum value of orientation difference is approximately 66.8 degrees (when the rotation axis does not coincide with a symmetry axis). This can be further explained this way: 1) orientation difference around the 4-fold axis can be mapped to [0, 45] degrees; 2) orientation difference around the 2-fold axes can be mapped to [0, 90] degrees, as similarly reflected by the angular distribution in the Supplementary Figure 3. These considerations will likely affect the range of the X axis of the Figure 5b. As nicely demonstrated in the Figure 5e, display of these particles with the D4 symmetrized map gives an impression that particles in the same clusters (identified by the authors) undertake somewhat similar orientations (especially the ones in Figure 5e-vi; the spiral arrangement). I support publication of the manuscript but please the authors further clarify this points in the figure legend or the method so that the readers are aware of these differences.

Reviewer #4

(Remarks to the Author)

All of our comments have been addressed. We support publication of this exciting work.

Version 3:

Reviewer comments:

Reviewer #1

(Remarks to the Author)

I thank the authors for further clarification and revision. I support publication of this manuscript in its current form.

Point-by-point responses to reviewers' comments

Reviewer #1 (Remarks to the Author):

In this manuscript, the authors characterized the cellular architecture of pyrenoid inside *Chlamydomonas* cells, a liquid-like compartment important for the efficiency of CO₂ fixation and for bioengineering applications. The authors performed cryo-electron tomography (ET) of cryo-focused ion beam milled *Chlamydomonas* cells, subtomogram averaging (STA) of Rubisco particles, molecular dynamics flexible fitting (MDFF) for the resulted EM map, as well as analysis of the Rubisco distribution within the cellular pyrenoid compartment, which is state of the art. Although the Rubisco complex appears as a special case for STA as they are abundant in the pyrenoid, I still feel that the methodology provided in this manuscript will advance the field by providing strategies for analyzing structures of relatively small cellular complexes like Rubisco (~13 nm), their structural variations, and their distribution profiles directly inside the cellular context.

I think there are a few points that could be further clarified:

1. In the manuscript, the authors selected 26 tilt series out of 66 (37+29) lamellae that are collected, considering the tomogram quality. It seems like the number of usable tilt series are considerably low, given the fact that normally at least about 5 tilt series per lamella can be collected with the settings described in the method section. I wonder if there are specific technical reasons that limit obtaining more tilt series of the pyrenoid regions from these lamellae. Or, is this due to a strict selection criterion of tomograms? If so, could the authors please provide supplementary figures or description in the text for the reasoning? This is of importance as such factors often relate to the final resolution of the structures obtainable by STA.

We appreciate the reviewer's comment. The lamella (~100-150 nm thick) arguably contains only 1-2% of the whole *Chlamydomonas* cell volume. As lamellae were made blindly, without CLEM, pyrenoid was not captured in every lamella, and in some lamellae, pyrenoid was only captured partially. For lamellae containing the pyrenoid, normally only one tilt series could be acquired for each pyrenoid as the neighbouring area was damaged by the beam during data collection. In addition, after the reconstruction of tomograms, we selected the ones with a mean residual error smaller than 1 nm during the reconstruction as these tomograms yielded more reliable results. We have now incorporated the description of tomogram selection into the revised manuscript, in the methods section.

2. In the STA pipeline, the authors classified the subtomogram particles into 20 initial classes. Out of these, the best class (of 17,713 particles with D4 symmetry applied) gave a resolution of 8 Å, with a pixel size of 1.97 to 1.98 Å used for data collection. Is there a possibility that some good particles were misclassified into the classes that they should not belong to? Could this lead to a relatively small percentage of particles contributing to the final high-resolution information of the EM map? If the authors have tried different number of classes to start with and found a number of 20 useful for this case, please include it in the description for the readers to understand the rationale.

Indeed, the choice of 20 classes was based on an extensive examination of different classification strategies, experimenting with various numbers of classes throughout the process, joining similar-looking classes, or sub-classifying heterogeneous-looking classes. The number of classes forms a compromise between resolution and heterogeneity, and a number of 20 classes was eventually chosen as it gave the highest resolution reconstructions for the dataset. A lower number (such as 10) apparently did not account for the vast heterogeneity, whereas a significantly larger number of

classes resulted in an insufficient number of particles per class and, thus, lower resolutions. The classification process was iterated in order to allow particles to migrate between classes.

Additionally, joining of the high-resolution class 12 with other classes did not improve resolution. Joining all the highest resolution classes did not improve resolution either. These indicate that the classes are distinct at the specified resolution, and Rubisco particles are highly heterogeneous, leading to a relatively small percentage of particles contributing to the final high-resolution map.

We have added the following text to the results section (Page 3):

“Additionally, we found that classifying into 20 classes yielded the best results. This likely represents a balance between the vast heterogeneity in the dataset and the number of images available per class. Attempts to classify into different numbers of classes, join, or sub-classify the resulting 20 classes, did not improve resolutions of the reconstructions.”

3. Regarding the stochastic distribution of Rubisco particles, did the authors profile the pair-wise distances and angles of different classes relative to each other, in order to reveal possible coordination of conformations of Rubisco within the pyrenoid? Also, some more discussion on the difference of Rubisco distribution within the pyrenoid from that in the α - and β -carboxysomes and possible factors contributing to this would be useful for understanding the general mechanism.

We thank the reviewer for this question. We have now included the analysis of pair-wise distances and angles of different classes relative to each other among the best-resolved classes: 7, 10, 12, and 13 in the revised manuscript (new **Figure 6**, new **Supplementary Fig. 8**). We also included more detailed discussions regarding the distribution of Rubiscos in pyrenoid compared to α - and β -carboxysomes in the revised manuscript (Page 10-11).

Minor points:

1. Line 56-57, “The active site is located at the interface between the C-terminal of one large subunit and the N-terminal of an adjacent one, such that two large subunits form a functional dimer harboring two active sites.” It would be useful to illustrate this with the resolved model/map in the figures.

We have revised the manuscript to include this suggestion. **Figure 2b** now includes a clearer view of the large subunit dimer interface and the two active sites. The upper and lower subunits are now colored light and dark blue respectively, the key segments from both active sites are colored dark red, and the 2CABP inhibitors are shown in black stick presentation.

2. Supplementary Fig. 9 (e-g), for superposition, could the author please use different colors for two structures? I also wonder if the pictures in (a) and (b) are consistent with the legend, as the N-ter model looks slightly better in (b) than that in (a), different from the description in the legend.

We have revised the colours of chains O and E presented in **Supplementary Fig. 9 (Supplementary Fig. 12** in the revised version). The legend for panel a and b has been modified to be consistent with the images.

3. Legend of Supplementary Fig. 12, “for (e) large and small subunits.”, (f) is missing in the text.

“(f)” is added.

Reviewer #2 (Remarks to the Author):

Summary

This manuscript presents the structural analysis of RuBisCOs within the pyrenoid of *Chlamydomonas reinhardtii* using cryo-electron tomography (cryo-ET) in combination with cryo-focused ion beam (cryo-FIB) milling, subtomogram averaging (STA), and molecular dynamics flexible fitting (MDFF). The authors revealed significant structural heterogeneity of RuBisCO complexes, including a closed (activated) conformation, local conformational changes at the active site, and interactions with binding proteins. These findings highlight the dynamic nature of RuBisCO in its native cellular environment. The study also explores the stochastic distribution of RuBisCO particles within the pyrenoid and the potential roles of binding proteins, particularly EPYC1. Overall, these findings make an important contribution to our understanding of RuBisCO in its native context within the pyrenoid. However, the following revisions would strengthen the manuscript.

Major comments:

1. The observation of structural heterogeneity of RuBisCO complexes within the pyrenoid is one of the most compelling results of this study. The authors do an excellent job of identifying different conformational states of RuBisCO (closed and open conformations) and providing insights into how this heterogeneity impacts RuBisCO's function. However, the manuscript could do more to quantify the extent of this heterogeneity and its implications for RuBisCO's catalytic activity. For instance, how do the authors identify different conformational states of RuBisCO at such rather low resolutions? Could you provide a quantitative summary of the relative abundances of different RuBisCO conformational states observed in your study? How does the distribution of these conformational states correlate with RuBisCO's enzymatic function within the pyrenoid? A more direct link to functional implications could strengthen this section.

We thank the reviewer for the excellent suggestions. One of the main observations in this study is the heterogeneity in Rubisco's functional states. We observe multiple classes, out of which, only a single class refined to high enough resolution, which enabled the identification of a closed conformational state (class 12). This was done by refining the Rubisco coordinated in the map density and comparison to known atomic structures of different functional states (Figure 2c). The rest of the classes were resolved at a resolution of worse than 10 Å, which does not allow us to determine their conformational states directly through the structure. Instead, we compared their densities to class 12, showing significant local variability at the active site and at the large subunit dimer interface. We interpret this finding as multiple functional conformations, however the exact functional state of the lower-resolution classes cannot be determined. Based on previous publications that examined conformational transitions at the Rubisco active site, the intermediate and the fully open conformations all comprise disordered loops, perhaps correlates with poorer resolution in classes other than 12. We conclude that these classes likely do not represent a closed state because they deviate from class 12 and that they are likely in an open or intermediate states. The proportions of the closed and other identified states are now added to the manuscript as detailed below. The observation of variable states, one of which being a closed state, correlates with a dynamic picture of Rubisco within the pyrenoid. Rather than purifying the enzyme and stabilizing in a single conformation, we captured a snapshot of live, functional cells. We observed that the in vivo Rubisco population consists of multiple functional states, each could represent a different stage of the catalytic cycle. We have made the following changed in the revised the manuscript:

1. The correlation between conformational variability, enzymatic function and kinetic traits is now discussed in a new paragraph added to the discussion (Pages 7-8).

2. We included additional descriptions of the variations in the lower-resolution classes in the results section (Page 5).
3. **Supplementary Table 3** now includes an additional column showing the proportion of each C1-refined class in the subtomogram dataset. The caption was modified accordingly.
4. The proportions of classes 7, 10, 12 and 13 have been added to **Figure 3a** and the caption was modified accordingly. We revised the results section to include the proportions of different classes.

2. The identification of potential binding proteins, specifically EPYC1, interacting with RuBisCO is another major contribution of this paper. While the interaction between RuBisCO and EPYC1 is compelling and well-supported by previous studies, the manuscript introduces an additional potential binding site at the interface between RuBisCO large subunit dimers. However, identification and functional validation of these binding proteins remains speculative. If this interaction involves an unknown protein, the authors could speculate about its identity or functionality by conducting a sequence homology search using known RuBisCO-binding partners. Alternatively, biochemical assays such as co-immunoprecipitation (Co-IP) or cross-linking mass spectrometry could be employed to identify the bona fide binding partners of RuBisCO in *Chlamydomonas reinhardtii*.

Following the reviewer's suggestion, we have analyzed the binding interfaces of some known Rubisco binding proteins from α and β -carboxysomes and compared these to the *in situ* pyrenoid Rubisco density map. This was done by overlaying the Rubisco coordinates from published high-resolution structures with our fitted MDFF model. As a result, we found a good match for the following Rubisco binding proteins, as illustrated in new **Supplementary Fig. 10**: CsoS2, a linker protein that mediates α -carboxysome formation (Oltrogge et al. 2020 *Nat Struct Mol Biol*), CcmM, which condensates Rubisco within β -carboxysomes (Wang et al. 2019 *Nature*), and CsoSCA, which is a carbonic anhydrase within the α -carboxysome (Blikstad et al. 2023 *PNAS*). The former two share a similar function with pyrenoid EPYC1, while the latter function correlates with the Cah3 carbonic anhydrase in *C. reinhardtii*. Cah3 was shown to localize to the pyrenoid, possibly near thylakoid tubules, but there is no direct evidence for interaction with Rubisco (Sinetova et al. 2012 *Biochimica et Biophysica Acta*, Blanco-Rivero et al. 2012 *PLOS ONE*).

Previous studies have looked at pyrenoid proteins using mass spectrometry and co-IP (Meyer et al. 2020 *Science advances*, Mackinder et al. 2016 *PNAS*, Mackinder et al. 2017 *Cell*, Nam et al. 2024 *Cell*), but evidence regarding this interface is vague. To draw conclusive evidence, it would require either a high-resolution *in situ* structure or purifying candidate proteins in complex with Rubisco and determining their structures using single-particle cryo-EM (as done in the case of EPYC1 in He et al. 2020 *Nature Plants*). We feel that both are beyond the scope of this study.

We therefore have added a new **Supplementary Fig. 10** showing that the binding site of Rubisco binding proteins from α and β -carboxysomes coincide with the extra densities at the large subunit dimer interface, and have modified the discussion accordingly (Page 8):

*“We observed evidence of potential Rubisco binding proteins primarily at two interfaces: adjacent to the small subunits and the at interface between large subunit dimers. The small subunit binding site has been shown to be the interface for the EPYC1 linker protein or other pyrenoid proteins which share a similar binding motif^{12,47}. The large subunit dimer interface has been shown to be the binding site of CsoS2 and CcmM linker proteins in α - or β -carboxysome Rubisco, respectively, as well as for the carbonic anhydrase enzyme, CsoSCA from α -carboxysomes. Interestingly, there is a good match between the binding location of these carboxysome proteins and the extra densities in our 3D classes (**Supplementary Fig. 10**). It is yet to be determined which homolog protein(s) bind to the pyrenoid Rubisco at the large subunit dimer interface.”*

“A third interface may be located next to the active site, as evident from the subtraction of class 12 from classes 7, 10, and 13 (Fig. 3b-d blue). This potential binding interface may be attributed to Rubisco activase, as observed in the cryo-EM structure of the complex between Rubisco and the cyanobacteria “Rubisco activase-like” protein⁴⁰. Alternatively, this may be the site for the interaction of neighboring Rubisco complexes within the pyrenoid.”

See more details regarding the latter discussion point below, Reviewer 4 comment 2.

3. The results showed that RuBisCO particles are distributed in a stochastic manner within the pyrenoid. It is interesting as it contrasts to the rigid, ordered packaging in carboxysomes. While the random distribution of RuBisCO particles is a significant and novel observation, the authors briefly mention the existence of local clusters of RuBisCO particles, but the functional significance of these clusters is not fully explored. The authors could enhance this section by conducting statistical analysis to further investigate the characteristics of these local clusters. For example, do these clusters somewhat resemble the spiral, fiber, or filamentous arrangements seen in carboxysomes? Whether the RuBisCO particles directly interact with each other within the clusters or could these interactions be mediated by EPYC1 or other binding proteins? Would you propose that there is a correlation between RuBisCO clustering and areas of higher CO₂ fixation activity in the pyrenoid?

We greatly appreciate the reviewer’s question, which helps to further improve the manuscript. In response, we have now conducted a detailed examination of local Rubisco classes and their spatial organization concerning CO₂ concentration (**new Figure 6a**). Our analysis confirms that the abundance of specific Rubisco classes within CO₂-rich regions differs substantially from the overall distribution (**new Figure 6a**). Additionally, we observed some short-range spiral and fiber-like arrangements in local clusters, although the majority of Rubisco complexes are packed in an amorphous arrangement (Revised **Figure 5e**, **Supplementary Fig. 8**).

4. The manuscript provides structural insights into the conformational changes around the active site of RuBisCO, including movements in the key loops (e.g., loop 6 and the N- and C-termini) that regulate substrate binding and catalysis. These findings are important for understanding RuBisCO's catalytic mechanism. However, given the relatively low resolution of the structural data, it is important to consider how confidently the authors can assign specific conformational states, particularly in the key loop regions of RuBisCO. More critical evaluations on the quality of the cryo-EM maps and fitting of the MDFF models should be provided.

We thank the reviewer for this suggestion. A new **Supplementary Fig. 16 & Supplementary Fig. 17** have been added with calculated Q-scores to quantify resolvability and matching between map density and fitted model. The quantifications indicate that the model is indeed reliable at the assigned resolution, particularly at the critical active site loops. The methods section is amended accordingly (Page 12). Furthermore, based on the Q-scores we conclude that the local agreement for the specific sites between the model and the density is well captured.

Furthermore, to enhance the novelty and impact of the manuscript, the authors could discuss how the conformational flexibility of these loops could affect the catalytic efficiency of RuBisCO under physiological conditions in response to varying CO₂ concentrations.

We thank the reviewer for this suggestion and have added the following text to the discussion (pages 7-8):

*“The work presents a snapshot of an active pyrenoid Rubisco population at variable stages of the catalytic turnover. Since *C. reinhardtii* Rubisco co-evolved with the pyrenoid, its kinetic parameters were adapted to the relatively high CO₂ concentration associated with the CO₂ concentrating*

mechanism (CCM). This mechanism promotes carboxylation and competitively inhibits oxygenation. Within the framework of kinetic trade-offs observed among Rubisco types, an association with the pyrenoid CCM generally promotes a higher carboxylation rate and weaker affinity for CO₂^{42,43}. The reaction catalyzed by Rubisco consists of distinct steps and intermediates of variable stability. Substantial conformational shifts and flexibility at the active site facilitate the catalytic cycle, including substrate binding, accommodation of the reaction intermediates, and substrate release^{10,19,46}. Additionally, conserved residues at the active site, such as the glycines and valine that maintain the hinge motion of loop 6, are linked to the underlying flexible mechanism as well as pyrenoid Rubisco's kinetic traits^{10,47}."

Minor comments:

1.The authors note the high level of structural heterogeneity in the dataset, which led to a highest resolution of 8.1 Å. The manuscript suggests that heterogeneity, including conformational variability and binding protein interactions, is a significant factor in reducing signal-to-noise ratio (SNR) and thus the achievable resolution. During data processing, while the authors used iterative 3D classification, more advanced techniques could be applied to handle heterogeneity more effectively. For example, multi-class averaging or focused refinement could help extract more accurate structural details from heterogeneous populations.

We thank the reviewer for this comment. We have experimented extensively with symmetry expansion and local refinements, but without much success. Particularly, we have tried expanding symmetry to both C4 and D4, and masking 2LS+2SS or 1LS+1SS accordingly during focused refinements. We have tried 3D classification without alignment or auto-refine with restricted local angular searches, as well as different binning. All resulted in ambiguous densities. We conclude that the SNR is too low for focusing on such small structures.

"Multi-class averaging" is part of the 3D classification algorithm in RELION, which we have used. Particles are allowed to migrate between classes according to the best match within each iteration. We normally set the number of 3D classification iterations to 25 or 30.

2.In the Methods section, the tilt series acquisition protocol is detailed and well-explained. However, the manuscript does not mention the rationale behind the tilt range (−12° to 12°) that. Tilt ranges can influence the final resolution, and a more explicit justification of these parameters would be helpful.

We apologize for the confusion and would like to clarify the data collection scheme. The tilt range in our data collection was set to ± 54° with an starting angle (pre-tilt angle in the manuscript) of ± 12°. The starting angle of ± 12° was set to compensate the pre-tilt from the cryo-FIB milling where the lamellae were made with a 12° angle to the grid surface. We have revised the Method section to be clear.

3.The methodology for particle picking and subtomogram averaging is well-described. However, the authors mention two rounds of 3D classification to discard "bad" subtomograms. The criteria for classifying "good" and "bad" subtomograms would benefit from further elaboration. How were "bad" subtomograms identified specifically (e.g., based on image quality, presence of distortions, lack of RuBisCO features)?

We thank the reviewer for this comment. We have now elaborated further in the methods section as follows:

“Good” 3D classes were selected based on the overall appearance of Rubisco features, the map’s resolution, and accuracy in the angular assignment. For example, 3D classes 4, 6, 8, 15 and 18 in Supplementary Fig. 2a, all feature different levels of distortions compared to the rest of the classes and were therefore considered as “bad”. We found that accuracy in the angular assignment (`_rlnAccuracyRotations`) to be a particularly indicative parameter for assessing the quality of 3D classes in 3D classification and 3D auto-refine jobs.

4. A deep learning-based method crYOLO was used for particle picking. It would be beneficial to provide more details on this method and how the network was trained. This will give the reader a better understanding of the network’s performance and its ability to accurately detect RuBisCO particles.

We have added a new Supplementary Figure (1) which shows the results of crYOLO particle picking.

We have added the following text to the methods section on crYOLO particle picking:

*Particle picking was done using the deep learning based crYOLO software (v1.9.3)⁵¹. The network was trained by manually annotating 12 sections from 3 different tomograms reconstructed in bin 2. Training was done using "crYOLO" architecture, low pass filtering frequency of 0.1 and 38-pixel boxes. CrYOLO provides a confidence value for each particle following prediction, and the user can adjust the confidence threshold for optimizing the picking results. We found that the default prediction parameters gave excellent results (confidence threshold 0.3. Tomography picking mode: tracing search range 25% of box, tracing memory 0 and tracing min length 5), resulting in 515,948 subtomograms from all tomograms (**Supplementary Fig. 1**)."*

5. Lines 81-82. The recently published paper describing the intact structure of α -carboxysome should be cited here (Nature Plants. 2024 Apr; 10(4):661-672).

Reference has been added as suggested.

6. Please provide all relevant cryo-EM maps for the reviewers to evaluate their qualities and the reliability of fitting of the models into the maps.

The maps are now available on the public data base EMDDB under the accession code EMD-52438 (class 12), other maps are included in the additional information of the same entry.

7. Figure 5 should be revised for more clarity. Currently, it does not obviously illustrate the arrangement of RuBisCO particles and their clustering in the pyrenoid.

We appreciate the reviewer’s suggestion. **Figure 5** is now revised for clarity, and a new **Supplementary Fig 8** is now included in the revised manuscript. For more clarity, Supplementary Figs. 6 and 7 are provided to demonstrate the distribution of Rubisco particles in different ranges.

Reviewer #3 (Remarks to the Author):

Reviewer #4 (Remarks to the Author):

Elad et al., present a detailed structural analysis of in situ sub-tomogram averages of the crucial CO₂ fixation protein Rubisco in the model green alga *Chlamydomonas reinhardtii*. They achieved the highest resolution structure of in situ Rubisco at 8.1 Å, which allowed previously published Rubisco coordinates to be refined within their map using Molecular Dynamics Flexible Fitting. The results support that the best resolved classes closely resemble the published open and closed Rubisco conformations, and demonstrate that Rubisco does not show large whole domain movements. Moreover, Rubisco is shown to display a large heterogeneity, which aside from being a result of the aforementioned open and closed conformation stems also from the presence of additional densities at the large subunit dimer-dimer interfaces and could potentially represent densities derived from yet unknown binding partners. The authors also investigate the spatial distribution of the different Rubisco states, and they do not find any compartmentalisation of different Rubisco classes. They are also able to corroborate previous results on Rubisco's nearest neighbour distances.

Whilst the findings are interesting, they predominantly are descriptive and give little new biological insight into pyrenoid function. Much of the data supports previous cryo-ET data performed on the *Chlamydomonas* pyrenoid. Including the detailed description of pyrenoid architecture provided in Engel et al. *eLife* (2015) and that Rubisco has a "liquid-like" organisation in the pyrenoid demonstrated by Freeman Rosenzweig et al. *Cell* (2017). Whilst identifying open and closed Rubisco states and additional densities on Rubisco provides advances there is a lack of biological insight or further experimental investigation. For example, it could have been interesting to look at Rubisco states between different relevant environmental conditions such as photoautotrophic growth at high and low CO₂. Whereas all the images are taken on mixotrophic grown cells where there is an unclear relationship between photosynthetic carbon fixation and acetate usage. The manuscript also fails to look at broader pyrenoid architecture, such as how the Rubisco matrix interfaces with the pyrenoid tubules or the starch sheath. The described work within a manuscript that looked at wider architecture would give better insights, although performing this on photoautotrophic grown cells would give more biologically relevant insights when the pyrenoid is fully operational within the CCM.

We sincerely appreciate the reviewer's comments. We share the interest in investigating the pyrenoid under various conditions; however, conducting such an analysis would require extensive experiments following the same workflow as in the current study, which unfortunately falls beyond its scope. That said, we have characterized the distribution of Rubisco in relation to its distance from the pyrenoid tubules, with the results now presented in new **Figure 6a**. As our primary goal was to elucidate the in-cell structure of native Rubisco, we specifically selected central regions devoid of the starch sheath for data collection. Consequently, at this stage, we are unable to analyze the relationship between native Rubisco and the starch sheath.

Reviewer knowledge gaps: The reviewers would like to indicate that assessing the MDFF modelling is outside the scope of their expertise.

Major comments:

1. The two paragraphs exploring the local variations between the four analysed classes are written very confusingly. For instance, "Interestingly, significant differences (density gain) occur between class 12 and all other classes at the active site (Fig. 3b-d blue)." This initially suggested that it is class 12 that has additional density compared to the other classes, when figure 3 shows the reverse to be true. I had the same issue with "As observed for the active site, class 12 shows major differences (density loss) at this interface compared to the other 3 classes (Fig. 3b-d, green)." It would be good to re-write this section to make it much clearer where the gain and loss is and relative to which classes.

We thank the reviewer for pointing out this ambiguity in the text. We hope that the modified text in these two sentences is clearer now (page 5):

“Interestingly, significant density gain is observed at the active when subtracting class 12 from each of the other three classes (7, 10 and 13), indicating that class 12 lacks density at this position relative to the three classes (Fig. 3b-d blue).”

“Significant conformational variability is seen also between the large subunit dimers near the 2-fold axis. Particularly, class 12 shows extra density at this interface compared to all three classes (Fig. 3b-d, green)”

2. Authors point out that the additional densities around the active site in classes 7, 10 and 13 most likely correspond to the mobility of the loop 6, N- and C-termini and loop 64-68 and relate to classes 7, 10 and 13 being in the open conformation, while class 12 is in the closed confirmation. This area is also where Rubisco activase is binding and performing its activity (Flecken et al., 2020). Could the authors comment whether any of the detected additional densities around the catalytic site could correspond to Rubisco activase?

We thank the reviewer for this comment. Indeed, we attempted to locate Rubisco activase (Rca) associated with Rubisco by expanding subvolume, but did not observe the hexameric Rca complex as it appears in Flecken et al., 2020. It is possible that Rca is less well structured, or interaction with Rubisco is more dynamic/flexible/heterogeneous, or the map resolution is not sufficient. Green algae Rca has also been shown to be less stable (Blayney et al. 2011, J. Am. Soc. Mass Spectrom.) compared to the Rsa-like complex in the Flecken et al., 2020 paper.

Instead, we observed a neighbouring Rubisco complex in a few 3D classes, and the linking density is positioned next to the active site, as shown in the image below, overlaid with the Rca-bound Rubisco model from Flecken et al., 2020 (PDB 6Z1F).

We have modified the following text (Page 5):

“The extra density can also be attributed to active site binding proteins such as Rubisco activase⁴⁰ or neighboring Rubisco complexes, which bind in close proximity to the active site.”

3. Authors show the presence of additional densities at the polar regions of C1 refined class 12 Rubisco, adjacent to the small subunit which they propose to be the EPYC1 protein bound at sub-stoichiometric concentration. Have the authors attempted symmetry expansion and local refinement to potentially increase the resolution of the region of interest?

We thank the reviewer for the comment. Please see the response to the reviewer #2 (minor comment 1).

4. The authors then take their particles from the four best resolved classes to assess their spatial organisation within the pyrenoid by mapping the pairwise distances and angles. In the text the authors state that the nearest neighbour analysis was of particles in class 12 only (line 246), but then in the figure the particle number is 215,272, meaning it contains the particles distributed in the 20 classes from the sub tomogram averaging. In this context it would be useful to understand how did the authors pick the “true” Rubiscos to be included in the analysis. Were these all crYOLO picks? As we are (and potentially other readers) not familiar with this software it would be useful to explain whether it provides scoring of picked particles and what was the thought process behind including these particular picks.

All particles included in the pairwise distances and angle analysis (n=215,272) were originally picked using crYOLO following by removal of “junk” using iterative classification in RELION. The STA structure of class 12 was used as the model for mapping back. We have now clarified this (Page 6 and **Figure 5a** legend).

The crYOLO deep learning software provides a confidence value for each particle following prediction, and the user can adjust the confidence threshold for optimizing the picking results. We found that the default confidence threshold (0.3) gave excellent picking results as exemplified in **Supplementary Fig. 1**. We have provided more details regarding the crYOLO picking procedure in the methods section (Page 10) and added new **Supplementary Fig. 1**.

5. The methods section indicates that the *Chlamydomonas* cells were grown mixotrophically in TAP media, similarly to the method used by Engel et al., and Freeman Rosenzweig et al. It appears that currently there is not much information on the impact of heterotrophic growth on photosynthesis under low CO₂, however isotope labelling has shown that even at 5% CO₂ as much as 22% of carbon was derived from acetate (Heifetz et al., 2000). In this context it is hard to conclude how generalisable the findings on the pyrenoid organisation are. Could authors elaborate on why did they choose to grow their cells mixotrophically and how generalisable the results are to photosynthesis by providing a comparison to TP grown cells or comparing the distribution of Rubisco at the growth conditions using TAP and TP media with fluorescent microscopy. Moreover, the fraction of Rubisco in the pyrenoid in synchronised cultures varies depending on the time in the day-night cycle (Mitchell et al., 2014), so could the authors specify at what point in the dark-light cycle did they vitrify the cells.

We thank the reviewer for the comments. We initially designed the experiment to decipher the in-cell structure of Rubiscos in the pyrenoid, thus followed the culturing conditions used in the previous cryo-ET work. As the reviewer pointed out, the different growth conditions could lead to different distribution of Rubiscos in the pyrenoid. We are interested in investigating this, and plan to conduct the experiments with both super-resolution fluorescence microscopy and cryo-ET, in future studies.

For the question regarding the dark-light cycle, we vitrified the cells when the cells were exposed to the light for 6 hours. We are also interested in bringing the dark-light cycle into our future experiments.

Minor comments:

78: “The difference in binding sites on Rubisco by these linker proteins is believed to be related to the packaging of Rubisco.” What is the support for this statement? Recent work on the *Chlorella* pyrenoid has shown that CsLinker (an EPYC1 analog) binds to the Rubisco large subunit (Barrett et al. 2024, *Nature Plants*).

We thank the reviewer for this comment. We have removed this statement from the introduction. We now discuss Rubisco packaging models and the involvement of linker proteins in more detail in the discussion (Page 8)

We do not observe extra density at the binding location of the CsLinker helices on the large subunit (as seen in the cryo-EM structure from Barrett et al. 2024) in any of our 3D classes. As mentioned in the Barrett et al. 2024 paper, although the *Chlorella* CsLinker is an EPYC1 homolog, their binding interfaces are distinct.

79-81: “As the packaging of Rubisco affects the utilization and assimilation of CO₂, understanding the in-situ organization of Rubisco particles has garnered significant attention” - Could the authors provide a reference for how different packing affects the utilization of assimilation of CO₂? Do the authors mean different packing between carboxysomes, pyrenoids and plants not containing CCM, or the impact of different packings in different pyrenoids?

We have deleted these statements from the introduction

81-84: “Recent studies have demonstrated intriguing packaging patterns for Rubisco in α - and β -carboxysomes which are confined by shells^{15,33,34}. Whether similar or distinguished packaging patterns exist for Rubisco in the pyrenoid remains elusive.”

Freeman Rosenzweig et al. 2017 demonstrate using cryo-ET that Rubisco packaging in the pyrenoid is liquid-like, with no clear organised packaging. The authors do discuss that their data agrees with this in the discussion, so it is a bit misleading to state it as remaining elusive.

We appreciate the reviewer’s comment. We acknowledge that Rubisco packaging in the pyrenoid is largely liquid-like as observed by Freeman Rosenzweig et al. 2017. However, we find that the Rubisco complexes show local short-range clusters, and therefore packing appears to be not completely random (**Figure 5 and Supplementary Fig. 8**). We have replaced the phrase “remains elusive” with “remains to be explored”.

98-99: “Moreover, the distribution of Rubisco in pyrenoid is revealed to be stochastic, different from that rigid packaging in the α - and β -carboxysomes.” The *Chlamydomonas* pyrenoid has been shown to display liquid-liquid phase separation. Is the finding of the stochastic distribution of Rubisco and its different classes something the authors consider novel and not expected given the LLPS?

We appreciate the reviewer’s comment. Indeed, the *Chlamydomonas* pyrenoid has been shown to display liquid-liquid phase separation, different from the rigid packaging in the α - and β -carboxysomes. We have carried out further analyses and found that there exist differences in the abundance of all Rubisco classes in CO₂-rich regions of the pyrenoid (new **Figure 6a**), and local short-range clustering of Rubisco particles (**Figure 5**). We have modified the manuscript accordingly.

119: The demonstration of heterogeneity of Rubisco within the pyrenoid is very interesting. Could the authors show the angular distribution of the four best classes to demonstrate that the difference in densities is not due to missing views?

We thank the reviewer for this comment. **Supplementary Fig. 3** now includes angular distribution and FSC plots for the four best-resolved 3D classes. The reconstructions show no preferred orientation that may result in density artifacts.

111-112: “To resolve the structure of Rubisco, deep learning-based particle picking was performed, and most discernible Rubisco particles were correctly picked (Fig. 1c).” –What stage of processing

are the particles from figure 1c derived from? Are these all crYOLO picks or are there the result of classification and “junk” removal?

The particles shown in **Figure 1c** are after the junk removal process. We have added a note to **Figure 1c** legend.

133: It would be helpful to mention that figure 2a is a result of MDFF docking when it is brought up first time (so line 133 not 137).

We thank the reviewer for this comment and have revised the manuscript accordingly (Page 4).

166: typo - Asp437 should be Asp473 (as per cited references).

This typo is corrected.

186: Can the authors include how prevalent each of the analysed classes is in the particle population.

We thank the reviewer for this comment. Supplementary Table 3 now includes an additional column listing the proportion of each class from the total C1-refined dataset (i.e. particles included in the “good” classes).

214-216: The authors also mention the densities at the large subunit dimer interfaces to be reminiscent of the areas of carboxysome matrix “and likely indicates the presence of Rubisco binding proteins within the pyrenoid.” The phrasing of “likely indicates the presence of Rubisco binding proteins within the pyrenoid” suggest as if the presence of Rubisco binding proteins was a novel discovery, despite the fact that we know of many proteins containing EPYC-like Rubisco binding motifs and of Rubisco-activase. It would be more appropriate to say it indicated the presence of yet-identified proteins likely possessing novel mode of Rubisco binding for Chlamydomonas, but identified in carboxysomes.

We thank the reviewer for this suggestion. We have revised the manuscript accordingly (Page 6).

249: “but slightly larger than the distance in β -carboxysomes (122 nm)” – typo, missing comma.

Corrected.

252: Can the authors specify what angle and distance constrains they used for the clustering?

We have now included a description of the angle and distance constraints in both the figure legend (**Figure 5e**) and methods section (Page 13)

278: “Consequently, we find no evidence for synchronization or compartmentalization of Rubisco dynamics” – I understand the lack of evidence of compartmentalisation, but I wonder if synchronisation is the right word here? Since the cells were grown in dark-light cycles, then in principle they should be synchronised in regards to their cell cycle. Therefore, if there is any temporal dependency of the Rubisco dynamics it could be that the differences would not be detectable? Different grids were plunged at different time during the day-night cycle, which is something that should be specified.

We agree with the reviewer and have corrected this sentence.

We confirm that all grids were plunged at the same time point of the day-night cycle and included the description in the methods section (Page 9).

The light intensity is given as lux, which does not relate to the Photosynthetic Active Radiation. Assuming that the lights used for growth were fluorescent lamps, then a rough calculation gives us 12,000 lux to be $\sim 162 \mu\text{mol m}^{-2} \text{s}^{-1}$ in photosynthetic photon flux density.

We thank the reviewer for providing this information. We followed the previous cryo-ET studies on *Chlamydomonas* and used their protocols. We did not use fluorescent lamps, but used white light.

Figures:

Figure 1: Nomenclature doesn't align with current field: Pyrenoid tubules are tubular thylakoids running through the pyrenoid matrix and minitubules are the small tubules found within the pyrenoid tubules that provide connections between the matrix and stroma.

We have now amended the naming accordingly.

Supplementary figure 2. Please label the FSC curves.

FSC curves are now labelled in the figure caption.

Figure 3: Add a not-lowpass filtered maps of the four best classes to the figure (currently only in the supplement).

We used not-lowpass filtered maps of the four best classes in Figure 3a.

References:

Most of the references in this paper are relevant with a few mistakes (see comments below), however in a few instances in the introduction the authors refer to multiple review papers instead of primary sources. Moreover, some of the introduction sentences contain multiple separate statements, that need references, but the authors often place them at the end of the sentence. This makes it difficult to locate which publication contains the relevant information, especially if they are broad review papers. Some examples include lines 46, 48, 66, 72.

We thank the reviewer for this suggestion. We have revised the references in the introduction to better support the statements.

42: "One-third of the global CO₂ is arguably fixed in algae by ribulose-1,5-bisphosphate carboxylase oxygenase, commonly known as Rubisco" – the paper says that 1/3 of photosynthesis is fixed by pyrenoids. Not all algae have pyrenoids, and some land plants such as hornworts do have pyrenoids. Moreover, the review is not the primary source of this information, but it is Mackinder et al. 2016 (referenced by ref5).

We corrected this mistake and moved the sentence to the third paragraph, which introduces the pyrenoids.

55: "Numerous structures of purified Rubisco have revealed the catalytic cycle in detail" - here a reference to a review discussing these Rubisco structures and the catalytic cycle would be appropriate

We have added two references:

Andersson, I. & Backlund, A. Structure and function of Rubisco. *Plant Physiology and Biochemistry* **46**, 275–291 (2008).

RuBisCO 3D structures - Proteopedia, life in 3D.
https://proteopedia.org/wiki/index.php/RuBisCO_3D_structures.

66: Reference 22 does not refer to Chlamydomonas.

This reference has been removed.

67: “The pyrenoid is primarily found in many eukaryotic algae and plays a crucial role in enhancing the efficiency of carbon fixation, a process believed to have been driven by the gradual decrease of atmospheric CO₂ over billions of years until the recent industrial revolution by humans^{21–25}.” (Elad et al., p. 2) – This is an example of the multiple statements that would be appropriate to reference separately, i.e. reference to where the pyrenoid is found and then a reference to the pyrenoid evolution.

References were split as suggested

78-79 – “The difference in binding sites on Rubisco by these linker proteins is believed to be related to the packaging of Rubisco.” It would be good to add relevant references to this sentence.

We have deleted this statement

82: the following reference also appears to be also relevant:
<https://doi.org/10.1016/j.str.2024.05.007>

This reference has been added.

399 – reference 50 refers to Rosetta and is inconsistent with the reference cited in line 198 (ref 37).

We think that there is a misunderstanding here.

The first reference is for ChimeraX software: Goddard, T. D. et al. UCSF ChimeraX: Meeting modern challenges in visualization and analysis. *Protein Science* 27, 14–25 (2018).

The second reference relates to the “1% false discovery rate threshold” method: Beckers, M., Jakobi, A. J. & Sachse, C. Thresholding of cryo-EM density maps by false discovery rate control. *IUCr* 6, 18–33 (2019).

Rosetta software is referenced elsewhere.

Reviewer #5 (Remarks to the Author):

Point-by-point responses to reviewers' comments

Reviewer #1 (Remarks to the Author):

Elad et al aimed to reveal the native structural arrangement and dynamics of Rubisco complex within a model system *Chlamydomonas reinhardtii*. In the revised manuscript, the authors clarified a few major points regarding the sample preparation, classification procedures in data processing, and post-processing analysis of distribution of Rubisco complexes. They mapped back the Rubisco into the tomograms and performed local clustering analysis for detailed illustration of potential higher-order assembly/coordination of Rubisco complexes. Also, the authors plotted the distribution of different classes of the complex in relation to the Pyrenoid tubule, as well as the relative distances/orientations of the best four classes based on the tomogram data, and concluded that the particles exhibit a predominantly stochastic distribution, with short-range local clustering of some particles in the pyrenoid. Regarding the binding proteins, the authors also performed further comparison and docking of previous models into their map and strengthened the point. Although the resolution of the maps remains insufficient to pinpoint the exact partner proteins, the study demonstrates the complexity of in-cell cryo-ET analysis, and future work with integrative strategies will hopefully help answer new questions brought out by the current manuscript.

I just have one remaining question that can be clarified for the pair-wise angle measurement. As the authors state the Class 12 model was used for the mapping-back analysis, please include in the method section whether or not the D4 symmetry has been included/considered during this analysis and if this could affect the angular distribution to some extent.

We thank the reviewer for the comment. All mapping back measurements were performed using results from asymmetric refinements. We have added the following sentences in the methods section to clarify this point:

“Particles included in this analysis (n=215,272) were originally picked using crYOLO, followed by the removal of “junk” using iterative classification without imposing symmetry in RELION4, as described above (Supplementary Table 3, Supplementary Fig. 2). Distance and angular measurements were performed using the C1-refined reconstructions, while the D4 symmetric class 12 map was used for display purposes only. “

Reviewer #2 (Remarks to the Author):

In the revised manuscript, the authors have addressed most major concerns raised in the initial review. The manuscript is now scientifically rigorous, clearly written, and increases our understanding of the assembly pattern of Rubiscos in the pyrenoid. The cryo-ET combined with cryo-FIB milling is really a powerful technique for investigating the structural dynamics of pyrenoid Rubiscos in the native environment. This work also paves the way for studying the structural and functional correlations of the pyrenoid Rubiscos for enhanced CO₂ assimilation. Therefore I support publication of this manuscript in its current form.

We appreciate the reviewer for the positive comments.

Reviewer #3 (Remarks to the Author):

Reviewer #4 (Remarks to the Author):

We would like to thank the authors for thoughtfully addressing our comments. The new provided figure 6 with classes in relation to tubules is interesting and a good addition to the manuscript. The authors have addressed all of the statements we thought were problematic and added a stronger emphasis on the structural aspect of the work. We think it is reasonable for them to say that getting more tomograms would be out of the scope of this study.

There are still some minor referencing issues with the last comment addressed:

Initial comment and rebuttal: "399 – reference 50 refers to Rosetta and is inconsistent with the reference cited in line 198 (ref 37).

Response: We think that there is a misunderstanding here.

The first reference is for ChimeraX software: Goddard, T. D. et al. UCSF ChimeraX: Meeting modern challenges in visualization and analysis. *Protein Science* 27, 14–25 (2018).

The second reference relates to the “1% false discovery rate threshold” method: Beckers, M., Jakobi,

A. J. & Sachse, C. Thresholding of cryo-EM density maps by false discovery rate control. *IUCrJ* 6, 18–33 (2019).

Rosetta software is referenced elsewhere."

We would like to clarify our initial comment but also highlight a referencing issue:

In line 466 (previously 399) the fragment "binarized at 1% false discovery rate threshold in ChimeraX (ref 50)" was initially a reference to Rohl, et al., 2004, which refers to Rosetta and not to the false discovery rate nor ChimeraX. However reference 50 is now: Zang, K., Wang, H., Hartl, F. U. & Hayer-Hartl, M. Scaffolding protein CcmM directs multiprotein phase separation in β -carboxysome biogenesis. *Nat Struct Mol Biol* 28, 909–922 (2021).

It appears that the references maybe out of sync and need to be checked.

We very much appreciate the reviewer's comments. We have corrected the reference number for ChimeraX (60), and the “1% false discovery rate threshold” method (38). Reference 50 cites the CcmM protein paper by Zang et al. We think references are correct now.

Point-by-point responses to reviewers' comments

Reviewer #1 (Remarks to the Author):

Reviewer #1 (Remarks to the Author):

I would like to thank the authors for clarifying the question. But I think the second part of the question has not been answered. As the authors pointed out, “the Distance and angular measurements were performed using the C1-refined reconstructions”. However, the D4 symmetry was applied for most structural analyses of this manuscript and thus a majority of results here have been interpreted with the symmetry applied, implying that the D4-symmetry is biologically meaningful. Based on the definition of D4 symmetry operation, the maximum value of orientation difference is approximately 66.8 degrees (when the rotation axis does not coincide with a symmetry axis). This can be further explained this way: 1) orientation difference around the 4-fold axis can be mapped to [0, 45] degrees; 2) orientation difference around the 2-fold axes can be mapped to [0, 90] degrees, as similarly reflected by the angular distribution in the Supplementary Figure 3. These considerations will likely affect the range of the X axis of the Figure 5b. As nicely demonstrated in the Figure 5e, display of these particles with the D4 symmetrized map gives an impression that particles in the same clusters (identified by the authors) undertake somewhat similar orientations (especially the ones in Figure 5e-vi; the spiral arrangement). I support publication of the manuscript but please the authors further clarify this points in the figure legend or the method so that the readers are aware of these differences.

We thank the reviewer for the comment.

While D4 symmetry was applied for much of the structural analyses of the complex in the paper, we did not apply D4 symmetry in the context of the tomogram, especially for template matching, particle orientation determination and particle distribution. The Pair-wise distance and angular measurements were performed using the C1 symmetry, such as shown in Figure 5, even though we used a D4 symmetrized Rubisco model for the placement. We have now clarified this point in the figure1 and figure 5 legends and also the method section.

Per reviewer's reference to Figure 5e, because the clustering analysis was conducted by thresholding the pair-wise distance and angle between neighbours as described in the legend and method, those particles indeed share the similar orientation as the pair-wise angle was set 0-30°.

Point-by-point responses to reviewers' comments

Reviewer #1 (Remarks to the Author):

I thank the authors for further clarification and revision. I support publication of this manuscript in its current form.

We thank the reviewer for the comment.